# A Neural Signal Codec with Resource Efficient Encoder for Implantable Brain Machine Interface Systems

## Abstract

In this paper, we present a neural signal codec (NSC) with a resource-efficient encoder for implantable brain machine interface (iBMI) systems. The proposed codec has a multiplication-free encoder with only 124-bit lightweight parameters, which is suitable for deployment at the edge of an iBMI system. To reduce the parameter size, a dynamic weight generation mechanism for parameter sharing within the window is implemented in the encoder design. On the decoder side of the codec, a conventional multilayer convolutional neural network with a specially designed loss factor – Energy Aware Loss (EAL) is adopted, which adds adaptive attention to the total loss function to improve reconstruction performance by emphasizing the signal energy intensive regions of the input data section. The parameter storage is reduced by 97% on the encoder side, compared to a conventional FC-based autoencoder with INT8-quantized weights. Large-scale evaluations show that NSC is capable of restoring high-fidelity neural signals and preserving the biological features across diverse neural signal datasets, making it a promising data compression approach for high-throughput iBMI systems. Furthermore, preliminary generalization experiments on other biomedical signals such as ECG (MIT-BIH) further demonstrate the potential of NSC as a general resource-efficient compression framework for streaming biosignals.

## 1 Introduction

In recent decades, the implantable brain machine interface (iBMI) system has become a research hot spot since it shows a promising potential to cure various neural-related diseases and to open a new gate for neuroscience research Musk & Neuralink (2019); Pollmann et al. (2024). A modern iBMI system typically consists of an implantable device and an external function module, such as a PC and robotics, as shown in Fig.1(a). The implantable device acquires the signal, typically a spike signal, from single neurons in the brain cortex, and transmits the acquired spike signals to the external function module through wireless communication. A critical design challenge in iBMI systems is to minimize the resource consumption of the implantable device, mainly dimension and power consumption, to achieve minimum surgery damage and long-term operation safety.

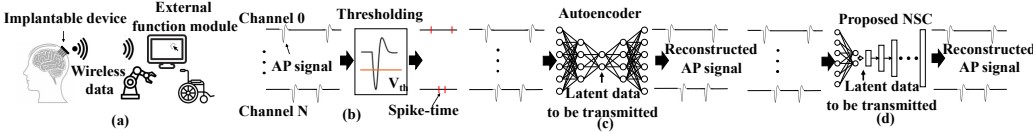

Figure 1: Diagram of (a) Typical iBMI system (b) Conventional Threshold Detection (c) Conventional AutoEncoder (d)The Proposed NSC

**High Throughput versus Resource Consumption.** High Throughput versus Resource Consumption. High throughput up to thousands of channels is required for an iBMI system to perform high-degree-of-freedom tasks such as virtual finger movement, 3D control of robotic arms, and complex control of e-Games Irwin et al. (2017); Willsey et al. (2025). However, high throughput would

result in high resource consumption. For example, a 1000-channel spike signal acquisition would generate 300M bits of raw data per second, which brings a heavy burden on power and device dimensions for the implantable device to handle and transmit. New emerging technologies such as Ultra-Wide-Band (UWB)Song et al. (2022); Ando et al. (2016) for high-speed wireless communication may help address this problem, but these technologies are still immature or unstable to be adopted in the iBMI system. Another alternative approach people normally use to address this throughput/resource dilemma is to compress the raw data for each individual channel before any further processing/transmitting. High loss compressing methods such as simple thresholding, spike detection Mukhopadhyay & Ray (1998) or on-chip sorting Valencia & Alimohammad (2021), are popularly used due to their simplicity and effectiveness in extracting critical spike-time information from the raw neural recordings, as shown in Fig.1(b), which significantly reduce data volume by only transmitting timestamps or sparse bi-nary indicators of spike events. However, they discard the morphological details of spike waveform, imposing strict requirements on the performance of downstream sorting or clustering algorithms on the external module. Low-loss compressing approaches such as Autoencoder and PCA Valencia et al. (2024) are also used to perform data compression as shown in Fig.1(c), which can effectively reduce the data size while keeping most information of the raw data. However, these algorithms are intensive on both computation and storage on the encoder side, which is impractical for an implantable device in an iBMI system.

Implantable neural interfaces demand compression codecs that are not only resource-efficient but also preserve task-critical information. We propose a Neural Signal Codec (NSC) Fig.1(d). With an encoder requiring only 124 bits of weight params and shift-add operations, two orders of magnitude fewer bits than models like PCA ($\sim 20k$ bits) or AE_FP32 ($\sim 16k$ bits). The key innovation is its targeted fidelity: while global waveform metrics (FULL-PSNR) are moderate, the NSC excels in preserving biologically crucial information. Our NSC achieves a 32:1 data compression ratio with a mean FULL PSNR value of more than 17.92, ROI PSNR value more than 19.67, with ROI waveform cluster F1 more than 0.96, ARI more than 0.85, and NMI more than 0.78 across all tested datasets. Saving 97% storage resource compared to the conventional INT8 weights quantized FC-based autoencoder design and dramatically reduces the required computational resource. The main contribution of this work is summarized as follows:

(i) We proposed an asymmetric encoder-decoder neural network architecture with a resource-efficient encoder that is suitable for lightweight edge deployment for high-fidelity data compression and reconstruction.

(ii) We constructed a resource-efficient encoder with a learnable, energy-aware windowing mechanism and shift/addition operation-based computation, optimized for an implantable device in an iBMI system.

(iii) We introduce a loss factor: Energy Aware Loss (EAL) factor, which adaptively updates the neural network weights in the training process based on the energy profile of the input spike signal, enabling an accurate and interpretable spike reconstruction.

The paper is organized as follows: Section 2 reviews related work on lightweight neural networks and on-chip neural signal compression. Section 3 describes the proposed NSC: encoder design, decoder architecture, and loss formulation. Section 4 details the experimental setup, datasets, and evaluation metrics, followed by quantitative results. Section 5 concludes the paper.

## 2 RELATED WORK

### 2.1 QUANTIZATION AND LIGHTWEIGHT NEURAL NETWORKS

Model compression techniques such as pruning and quantization are widely used to reduce the computation and memory footprint of neural networks for deployment in resource-constrained environments (e.g., mobile or embedded systems). Early pruning methods removed redundant weights post-training LeCun et al. (1989), and structured strategies like channel pruning Li et al. (2017). Quantization reduces bit-widths of weights and activations, with early approaches including fixed-point training Lin et al. (2016), BinaryConnect Courbariaux et al. (2016a), and XNOR-Net Rastegari et al. (2016). More recent practices like quantization-aware training (QAT) and post-training quantization (PTQ) better preserve model accuracy under low precision Jacob et al. (2017).

Within the Transformer and LLM domain, ZeroQuant Yao et al. (2022) enables group-wise quantization and applies layer-wise knowledge distillation (LKD) to retain accuracy. SmoothQuant Xiao et al. (2024) improves activation quantization by redistributing outliers into weights. On the extreme end, BitNet Wang et al. (2023) proposes BitLinear, a ternary-weighted alternative to standard linear layers. PB-LLM Shang et al. (2023) adopts a mixed-precision strategy, binarizing most weights while keeping key ones in higher precision. For joint pruning and quantization, Bayesian Bits van Baalen et al. (2020) learns both sparsity and bit-widths during training.

## 2.2 ON-CHIP NEURAL SIGNAL COMPRESSION

On-chip neural signal compression is key to reducing transmission bandwidth in iBMI systems.

High-loss methods focus on event detection and on-chip sorting. For example, Kim et al. (2019); Hwang et al. (2025) transmit only spike timestamps using event-driven compression. On-chip sorting approaches like Chen et al. (2023); Han et al. (2025) employ OSort-inspired pipelines, while Binarized Neural Network (BNN) based classifiers in Valencia & Alimohammad (2021) provides an effective low-power spike classification.

Low-loss methods aim to reconstruct spike waveforms. Compressed sensing approaches appear in Liu et al. (2016), and PCA-based real-time compression is reported in Lemaire et al. (2022). NNs like undercomplete autoencoders are also used for low-power hardware compression in Thies & Alimohammad (2019); Valencia et al. (2024), and other methods perform segmentation and pruning of low-importance waveform regions (Guo et al. (2023)). And the work Liu et al. (2024) uses ConvSNN utilize spike-oriented convolution data flow.

## 3 METHODOLOGY

### 3.1 MOTIVATION AND PROBLEM DEFINITION

Neural spike waveforms are high-bandwidth yet highly structured signals. Transmitting them in raw form from an iBMI device is prohibitive due to extreme limits on bandwidth, energy, and storage. This naturally motivates a representation learning problem: learn a compact latent code that preserves task-relevant fidelity while remaining feasible under strict hardware constraints.

Formally, we aim to design an encoder–decoder pair, where only the encoder is deployed on-chip. The encoder $f_\theta : \mathbb{R}^T \to \mathbb{Z}^{T'}$ maps an input spike waveform $\mathbf{x}$ into a discrete latent code $\mathbf{z}$, which is transmitted off-chip. The decoder $g_\phi : \mathbb{Z}^{T'} \to \mathbb{R}^T$ runs externally to reconstruct $\hat{\mathbf{x}} = g_\phi(\mathbf{z})$.

Unlike conventional autoencoders that optimize solely for reconstruction, our encoder must also satisfy strict resource constraints: (i) *bit-rate*, limited by the maximum transmission rate $R_{\max}$; (ii) *parameter storage*, constrained by $P_{\max}$; and (iii) *compute budget*, bounded by $C_{\max}$. Thus the learning problem is to minimize reconstruction error while ensuring that the quantized code length, parameter footprint, and operations of $f_\theta$ remain within hardware budgets:

$$\min_\theta \ \mathbb{E}_{\mathbf{x} \sim \mathcal{D}} \big[ \| \mathbf{x} - g_\phi(f_\theta(\mathbf{x})) \|_2^2 \big] \tag{1}$$

$$\text{s.t.} \quad \mathbb{E}_{\mathbf{x}}[L(f_\theta(\mathbf{x}))] \cdot N \cdot f_s \leq R_{\max}, \ \|\theta\|_0 \leq P_{\max}, \ \text{Ops}(f_\theta) \leq C_{\max}.$$

Here $L(\cdot)$ denotes the code length in bits, $N$ the number of channels, and $f_s$ the sampling rate. We use $\text{Ops}(\cdot)$ instead of general MACs to emphasize that only *shift-and-add* operations are allowed in our encoder. This formulation highlights a distinctive challenge for representation learning. Unlike conventional autoencoders that optimize solely for fidelity, our encoder must simultaneously ensure resource efficiency under hardware-level constraints. It therefore combines machine learning objectives with physical implementation feasibility.

### 3.2 NSC ENCODER DESIGN

The Neural Signal Codec (NSC) encoder compresses an aligned input window into a low-bit integer code, designed to be efficiently implementable on hardware using only *shift-and-add* operations,

without relying on general-purpose multipliers or floating-point units. The encoder design leverages three key ideas: (i) a nonlinear energy operator (NEO) to highlight spike regions, (ii) compact, shared learnable parameters that generate discrete per-sample shift factors, and (iii) quantization of all parameters to 4 bits.

**Input and NEO.** Given an input waveform $\mathbf{x} \in \mathbb{Z}^T$, we can obtain its nonlinear energy operator (NEO) value from the upstream detection module, which is not included in our compression process:

$$e_n = x_n^2 - x_{n-1}x_{n+1}, \qquad 2 \le n \le T-1. \tag{2}$$

Afterwards, we perform logarithmic operations on NEO with :

$$\text{LNE}_n = \lfloor \log_2(\max(e_n, 1)) \rfloor, \tag{3}$$

This transformation serves several purposes. The original NEO values are 16-bit, which are much larger in magnitude than the 8-bit input data. By clamping the minimum to 1 before taking the logarithm, we suppress contributions from very low values, which typically correspond to noise. Since the NEO reflects the local signal energy, smaller values are often associated with background noise. After this logarithmic mapping, the result provides a compact integer representation of local signal energy that is well-suited for subsequent shift-based scaling operations.

**Window Partition and Parameters.** The input $\mathbf{x}$ of length $T$ is partitioned into $w$ windows with differentiable boundaries $0 = b_0 < b_1 < \cdots < b_w = 1$. The boundaries are computed by a cascade of sigmoids applied to learnable parameters $\mathbf{p} \in \mathbb{R}^{w-1}$. Let $s_i = \sigma(p_i)$, where $\sigma(\cdot)$ denotes the sigmoid function:

$$b_0 = 0, \quad b_i = b_{i-1} + (1 - b_{i-1}) \cdot s_{i-1}, \quad i = 1, \ldots, w, \quad b_w = 1$$

The actual boundary positions in samples are $B_i = b_i \cdot T$. This formulation ensures $b_i \in (b_{i-1}, 1)$ and enables differentiable, trainable windows whose sizes adapt during training. The boundaries are initialized to equal divisions of the input.

Let the latent dimension be $d$. The total parameter sizes are $\alpha, \beta \in \mathbb{R}^{w \times d}$ and $\gamma \in \mathbb{R}^w$. For each window $i$ and latent dimension $j$, the shared learnable parameters $\alpha_{i,j}$ and $\beta_{i,j}$ produce discrete per-sample shift factors. Additionally, each window $i$ has a scaling factor $\gamma_i$. All parameters are quantized to 4 bits for inference.

**Quantization and Hardware Mapping.** Parameters $\alpha, \beta, \gamma$ are quantized to 4 bits, and the straight-through estimator (STE) is used during training. The quantized parameters are:

$$\alpha_{q,i,j} \in \left\{0, \tfrac{1}{16}, \ldots, \tfrac{15}{16}\right\}, \quad \beta_{q,i,j} \in \{-8, \ldots, 7\}, \quad \gamma_{q,i} \in \left\{0, \tfrac{1}{16}, \ldots, \tfrac{15}{16}\right\}.$$

This ensures that inference-time computations reduce to integer shifts and additions, fully eliminating multiplications. All the $m/16$ process can be achieved by shift-add-shift operation. With $w = 3$ windows and latent dimension $d = 4$, the total parameter bit budget is

$$3 \cdot 4 \cdot (4 + 4) \text{ bits} + 3 \cdot 4 \text{ bits} + 2 \cdot 8 \text{ bits} = 124 \text{ bits},$$

with 8-bit boundaries storage for hardware alignment.

**Forward Pipeline.** This process can also seen in Figure 2. For each sample $n$ in window $i$ and $j$-th latent dim ($n \in \{b_{i-1}, b_{i-1} + 1, \ldots b_i - 1\}, j \in \{0, 1, \ldots, d-1\}$), let $x_n^{(i)}$ and $e_n^{(i)}$ represent the $i$ window segment of full $x_n$ and $e_n$, the forward computation is

$$\text{shift}_{j,n}^{(i)} = \lfloor \alpha_{q,i,j} \cdot \text{LNE}_n^{(i)} + \beta_{q,i,j} \rfloor, \tag{4}$$

$$\text{scale}_{j,n}^{(i)} = 2^{\text{clamp}(\text{shift}_{j,n}^{(i)} - 8, -8, 7)}, \tag{5}$$

$$y_{j,n}^{(i)} = \lfloor x_n^{(i)} \cdot \text{scale}_{j,n}^{(i)} \rfloor. \tag{6}$$

Aggregating over the samples in each window gives a window-wise sum $y_j^{(i)}$, which is then scaled by the window factor $\gamma_{q,i}$. The final compressed latent code is

$$\mathbf{z}_j = \sum_{i=1}^{w} \mathbf{z}_j^{(i)} = \sum_{i=1}^{w} \lfloor \gamma_{q,i} \cdot y_j^{(i)} \rfloor, \quad \mathbf{z} \in \mathbb{Z}^{1 \times d}. \tag{7}$$

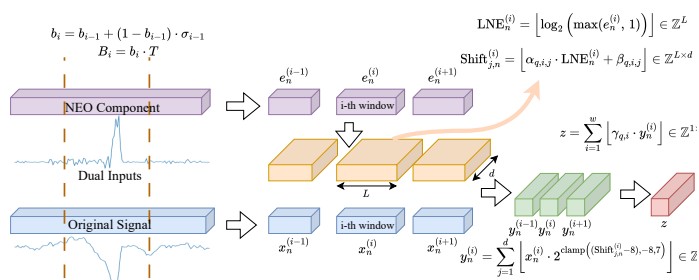

Figure 2: NSC Encoder Forward Pipeline.

### 3.3 DECODER ARCHITECTURE

The decoder operates entirely off-chip and is thus free from hardware constraints. We adopt a flexible design consisting of a channel-expansion layer, three symmetric downsampling and upsampling blocks with residual connections, and a final output projection. The compressed code $\mathbf{z} \in \mathbb{Z}$ is first expanded into a 1D feature map; the downsampling path increases channel capacity while reducing temporal resolution, and the upsampling path restores the waveform length. Each block uses Conv1d or ConvTranspose1d layers followed by residual Conv–BN–GELU modules, and the final projection employs two Conv–BN–GELU layers and a Conv1d layer to output the reconstructed signal.

### 3.4 ENERGY-AWARE LOSS (EAL)

Standard reconstruction losses weight all time points equally, even though only a small temporal region around the spike event is critical for accurate recovery. To better align optimization with the intrinsic structure of neural waveforms, we introduce the *Energy-Aware Loss* (EAL), which adaptively emphasizes high-energy regions identified by the nonlinear energy operator (NEO).

**Energy-based weighting.** For each input waveform $\mathbf{x}^{(b)} \in \mathbb{R}^T$ in the batch ($b = 1, \ldots, B$) with corresponding NEO $\mathbf{e}^{(b)}$, we first smooth $\mathbf{e}^{(b)}$ to suppress spurious fluctuations, yielding $\tilde{\mathbf{e}}^{(b)}$. A normalized weight distribution is then derived:

$$W_n^{(b)} = \frac{\mathcal{F}(\tilde{e}_n^{(b)})}{\sum_{m=1}^{T} \mathcal{F}(\tilde{e}_m^{(b)})}, \qquad \sum_{n=1}^{T} W_n^{(b)} = 1, \tag{8}$$

where $\mathcal{F}$ denotes a smoothing kernel (e.g., Gaussian or Laplace). This distribution serves as a soft attention mask, concentrating weight near the spike region. Given decoder reconstruction $\hat{\mathbf{x}}^{(b)}$, the loss is defined as

$$\mathcal{L}_{\text{EAL}} = \frac{1}{B} \sum_{b=1}^{B} \sum_{n=1}^{T} W_n^{(b)} \, \ell\big(x_n^{(b)}, \hat{x}_n^{(b)}\big), \tag{9}$$

where $\ell$ is a pointwise reconstruction cost (we adopt MSE). Unlike fixed windows or hard spike alignment strategies, EAL adapts continuously to each waveform's energy distribution. This allows the model to focus capacity on the informative spike region while naturally down-weighting background noise, improving reconstruction fidelity where it matters most.

## 4 EXPERIMENTS

### 4.1 DATASETS AND SETTINGS

**Baselines** We compare our designed NSC with several representative baselines. (i) *AE_FP32*, standard FC based AE. (ii) *AE_INT8* with 8-bit quantized-weights. (iii) *AE_INT1.4* uses a Hardtanh activation function with output precision aligned to Q2.8 and weight precision to Q1.4, where QM.N

represents a fixed-point representation with M bits for the integer part and N bits for the fractional part Valencia et al. (2024). (iv) *PCA* (Principal Components Analysis): A classical linear dimensionality reduction baseline for lossy signal compression Lemaire et al. (2022). (v) *CS* (Compressed Sensing): A sparsity-driven method that projects signals through a random sensing matrix and reconstructs them via sparse recovery Donoho (2006). (vi) *VQ-VAE*: A neural generative model with discrete latent variables using vector quantization for compact learning van den Oord et al. (2017). (vii) *BNN*: binary neural network with binary weights Courbariaux et al. (2016b).

**Datasets** We evaluate our method on one synthesis datasets and multiple real recording datasets, including neuronal, brain region and other recordings. (i) Quiroga (QU) Quiroga (2020), a standard datasets that has been widely used in the evaluation of spike-sorting. Generated by adding spike waveform templates to background noise of various levels. (ii) Ganglion Cells (GC) Spampinato et al. (2018), with extracellular recordings ground truth from simultaneous juxtacellular signals with 256 channels. (iii) Hippocampal (hc1) Henze et al. (2000), include the CA1 extracellular recordings with spike ground truth. (iv) Neuropixels (NP) Steinmetz et al. (2024), recorded from visual cortex, hippocampus, and some parts of thalamus with 384 channels. (v) MIT-BIH Arrhythmia (MIT-BIH) Goldberger et al. (2000), which is a standard collection of two-channel ECG recordings used for arrhythmia research.

**Data processing** All datasets were preprocessed by extracting signal segments containing spike events, with $T = 128$ samples (4ms data in 32kS/s), with spikes aligned to the midpoint of the window. The input data was quantized to 8-bit signed integers, and the corresponding NEO values were computed in advance. Both were stored in standardized npy files for convenient access. We focus on evaluating the region of interest (ROI)—the central 64-point segment, 2ms data window in 32kS/s, which contains the spike event. Details can be found in Appendix.

**Settings and Evaluations** All ablation studies and experiments are conducted with five fixed seeds **1, 2, 3, 4, 5** and setting latent dim of 4 and input size of 128 ($\times 32$ data compression). Results are reported as the mean±std (standard deviation) across all seeds. For reconstruction quality, we evaluate using PSNR, SNDR, and NRMSE, where higher PSNR and SNDR indicate better performance, and lower NRMSE is better. Where PSNR penalizes pixel-wise errors, SNDR captures noise and distortion, and NRMSE provides normalized error scaling. And for the downstream evaluation, we choose the simple K-means algorithm evaluated with three metrics, including F1, ARI, and NMI. F1 is the harmonic mean of precision and recall, ARI measures chance-corrected clustering agreement, and NMI quantifies normalized mutual information between label assignments. All models were trained for 100 epochs with 7:1:2 random division for train/valid/test. Validated by the best PSNR on the ROI. AdamW optimizer was used with a learning rate of $1 \times 10^{-3}$, additional weight decay of $1 \times 10^{-4}$ is applied to all non-quantized parameters. We also applied gradient norm clipping with a maximum of 10 to ensure training stability. In the comparison part, all baseline models were trained with MSE loss. All done in a single RTX 5090 GPU, Xeon 8470Q CPU, with Python 3.12.3 and torch version of 2.8.0+cu128. Details can be found in Appendix.

## 4.2 ABLATION STUDIES

We conduct a series of ablation studies to evaluate the influence of design choices on our NSC and EAL loss. The base setting adopts our proposed encoder with all parameters in full precision, a fixed three-window segmentation, and mean squared error (MSE) as the training loss. Here, we choose QU Difficult1Noise02 (QU D1N2) and NP channel 1 (NP channel 1) for the ablation study. Results are shown in Tables 1.

**Quantization strategies.** We compare full precision (FP), quantization-aware training (QAT), and post-training quantization (PTQ). QAT introduces a noticeable performance drop relative to FP, while PTQ completely fails (negative PSNR and SNDR). The failure of PTQ is mainly due to our encoder's dynamic scaling, which depends on the exponentiation of weight terms. Direct uniform quantization breaks the continuity of these exponentials and leads to large reconstruction instability. This suggests that QAT is necessary to maintain encoder functionality under quantization.

**Window mechanisms.** We study the effect of learnable window boundaries and of varying the number of windows $w$. Making the window boundaries learnable (wlr) produces very similar results to the fixed case, indicating that the mean division initialization is already near-optimal. As for the number of windows, the general trend across datasets is that performance improves with larger $w$ (up

to 4 or 5), consistent with finer local adaptation. However, we adopt $w = 3$ as our default based on two considerations: (i) it matches a physical prior of compressed signal waveform (LFP–spike–LFP (local field potential)). (ii) The encoder parameter count grows approximately linearly with $w$, larger window counts increase storage cost without consistent gains across datasets.

Table 1: Design Ablation study on synthetic dataset (QU D1N2) and real dataset (NP ch1).

| Setting | Syn (QU D1N2) | | |
| --- | --- | --- | --- |
| | $PSNR_{FULL/ROI} \uparrow$ | $SNDR_{FULL/ROI} \uparrow$ | $NRMSE_{FULL/ROI} \downarrow$ |
| Base | $18.50 \pm 0.36 / 18.37 \pm 0.54$ | $3.32 \pm 0.35 / 4.85 \pm 0.51$ | $0.13 \pm 0.01 / 0.13 \pm 0.01$ |
| QAT | $15.48 \pm 1.01 / 16.83 \pm 1.06$ | $0.30 \pm 1.02 / 3.31 \pm 1.06$ | $0.17 \pm 0.02 / 0.15 \pm 0.02$ |
| PTQ | $N/A / N/A$ | $N/A / N/A$ | $N/A / N/A$ |
| wlr | $18.43 \pm 0.28 / 18.46 \pm 0.66$ | $3.26 \pm 0.28 / 4.94 \pm 0.67$ | $0.13 \pm 0.01 / 0.13 \pm 0.02$ |
| $w = 1$ | $17.61 \pm 0.21 / 17.71 \pm 0.48$ | $2.44 \pm 0.21 / 4.19 \pm 0.46$ | $0.14 \pm 0.00 / 0.14 \pm 0.01$ |
| $w = 2$ | $18.24 \pm 0.38 / 18.54 \pm 0.48$ | $3.07 \pm 0.38 / 5.02 \pm 0.51$ | $0.13 \pm 0.00 / 0.13 \pm 0.01$ |
| $w = 4$ | $18.85 \pm 0.27 / 18.83 \pm 0.13$ | $3.67 \pm 0.28 / 5.31 \pm 0.14$ | $0.12 \pm 0.01 / 0.12 \pm 0.00$ |
| $w = 5$ | $18.39 \pm 0.65 / 19.17 \pm 0.70$ | $3.21 \pm 0.66 / 5.65 \pm 0.66$ | $0.13 \pm 0.01 / 0.12 \pm 0.01$ |
| Setting | Real (NP ch1) | | |
| | $PSNR_{FULL/ROI} \uparrow$ | $SNDR_{FULL/ROI} \uparrow$ | $NRMSE_{FULL/ROI} \downarrow$ |
| Base | $17.45 \pm 0.39 / 16.94 \pm 0.35$ | $2.27 \pm 0.38 / 3.67 \pm 0.33$ | $0.14 \pm 0.01 / 0.15 \pm 0.01$ |
| QAT | $15.63 \pm 0.80 / 15.22 \pm 1.40$ | $0.45 \pm 0.80 / 1.95 \pm 1.38$ | $0.18 \pm 0.02 / 0.19 \pm 0.03$ |
| PTQ | $N/A / N/A$ | $N/A / N/A$ | $N/A / N/A$ |
| wlr | $17.36 \pm 0.30 / 16.87 \pm 0.37$ | $2.17 \pm 0.30 / 3.60 \pm 0.35$ | $0.14 \pm 0.00 / 0.15 \pm 0.01$ |
| $w = 1$ | $17.04 \pm 0.29 / 16.31 \pm 0.23$ | $1.86 \pm 0.27 / 3.04 \pm 0.22$ | $0.15 \pm 0.01 / 0.17 \pm 0.01$ |
| $w = 2$ | $17.54 \pm 0.33 / 16.76 \pm 0.36$ | $2.36 \pm 0.28 / 3.49 \pm 0.35$ | $0.14 \pm 0.01 / 0.16 \pm 0.01$ |
| $w = 4$ | $17.90 \pm 0.30 / 17.49 \pm 0.28$ | $2.72 \pm 0.26 / 4.22 \pm 0.30$ | $0.13 \pm 0.00 / 0.14 \pm 0.00$ |
| $w = 5$ | $17.54 \pm 0.51 / 17.40 \pm 0.55$ | $2.36 \pm 0.47 / 4.13 \pm 0.56$ | $0.14 \pm 0.01 / 0.15 \pm 0.01$ |

· Mean $\pm$ std over five seeds.

**Loss Functions** We further investigate the effect of different loss formulations on NSC training. We also evaluate our proposed Energy-Aware Loss (EAL) in several variants: vanilla (direct neo guided), Laplace, Gaussian, and Cauchy smoothing schemes. The purpose of EAL is to emphasize signal regions with high energy, so as to improve fidelity in the regions of biological importance even at the cost of slightly reduced global metrics. Results are summarized in Table 2.

On the synthetic datasets, the vanilla EAL-vanilla actually reduces both FULL and ROI PSNR relative to plain MSE. By contrast, distributional EAL variants consistently increase ROI PSNR while sacrificing some FULL-PSNR. This demonstrates that coupling energy-aware reweighting with an explicit distributional form effectively concentrates model capacity on spike regions.

On the real datasets, all EAL variants yield small but consistent ROI gains over plain MSE. We additionally evaluate a combined setting (wlr + EAL-Gaussian) for NP with learnable windows together with EAL-Gaussian, the best variant. Showing that adaptive windowing helps when spike widths vary in real recordings.

Table 2: Loss Ablation study on synthetic datasets (QU D1N2) and real datasets (NP ch1).

| Setting | Syn (QU D1N2) | | |
| --- | --- | --- | --- |
| | $PSNR_{FULL/ROI} \uparrow$ | $SNDR_{FULL/ROI} \uparrow$ | $NRMSE_{FULL/ROI} \downarrow$ |
| Base | $\mathbf{18.50 \pm 0.36} / 18.37 \pm 0.54$ | $\mathbf{3.32 \pm 0.35} / 4.85 \pm 0.51$ | $\mathbf{0.13 \pm 0.01} / 0.13 \pm 0.01$ |
| EAL-vanilla | $17.43 \pm 0.82 / 17.63 \pm 0.23$ | $2.25 \pm 0.81 / 4.10 \pm 0.24$ | $0.14 \pm 0.01 / 0.14 \pm 0.01$ |
| EAL-Laplace | $17.90 \pm 0.51 / 19.38 \pm 0.11$ | $2.72 \pm 0.51 / 5.86 \pm 0.09$ | $0.18 \pm 0.05 / 0.11 \pm 0.00$ |
| EAL-Gaussian | $17.96 \pm 1.37 / \mathbf{19.48 \pm 0.14}$ | $2.78 \pm 1.36 / \mathbf{5.95 \pm 0.12}$ | $0.14 \pm 0.03 / 0.11 \pm 0.00$ |
| EAL-Cauchy | $17.44 \pm 1.63 / 19.43 \pm 0.09$ | $2.26 \pm 1.63 / 5.91 \pm 0.09$ | $0.15 \pm 0.03 / 0.11 \pm 0.00$ |
| Setting | Real (NP ch1) | | |
| | $PSNR_{FULL/ROI} \uparrow$ | $SNDR_{FULL/ROI} \uparrow$ | $NRMSE_{FULL/ROI} \downarrow$ |
| Base | $17.45 \pm 0.39 / 16.94 \pm 0.35$ | $2.27 \pm 0.38 / 3.67 \pm 0.33$ | $0.14 \pm 0.01 / 0.15 \pm 0.01$ |
| EAL-vanilla | $17.44 \pm 0.32 / 17.03 \pm 0.28$ | $2.26 \pm 0.28 / 3.76 \pm 0.26$ | $0.14 \pm 0.00 / 0.15 \pm 0.01$ |
| EAL-Laplace | $17.45 \pm 0.36 / 17.77 \pm 0.17$ | $2.27 \pm 0.37 / 4.50 \pm 0.16$ | $0.15 \pm 0.01 / 0.14 \pm 0.00$ |
| EAL-Gaussian | $17.51 \pm 0.60 / \mathbf{17.79 \pm 0.17}$ | $2.33 \pm 0.63 / \mathbf{4.52 \pm 0.15}$ | $0.14 \pm 0.01 / 0.14 \pm 0.00$ |
| EAL-Cauchy | $17.47 \pm 0.31 / 17.77 \pm 0.14$ | $2.29 \pm 0.33 / 4.50 \pm 0.13$ | $0.14 \pm 0.01 / 0.14 \pm 0.00$ |
| wlr + EAL-G | $\mathbf{17.65 \pm 0.20} / 17.76 \pm 0.17$ | $\mathbf{2.47 \pm 0.16} / 4.49 \pm 0.16$ | $0.15 \pm 0.00 / 0.14 \pm 0.00$ |

· Mean $\pm$ std over five seeds.

On the QU D1N2 dataset, the loss curves reveal several consistent trends (Fig. 3). Compared to plain MSE, the vanilla EAL exhibits faster convergence and reaches a stable plateau within fewer epochs. Both MSE and EAL show nearly overlapping training and validation curves, suggesting

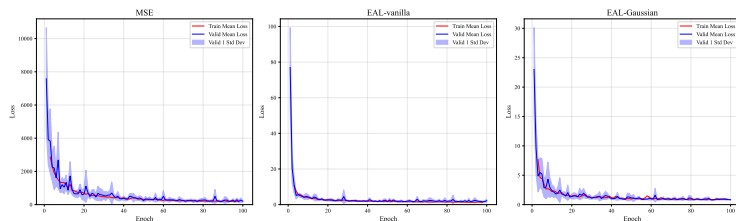

Figure 3: Training (red) and validation (blue) loss curves on QU D1N2.

that the introduction of EAL does not increase the risk of overfitting. Among the EAL variants, the Gaussian weighting achieves the lowest final validation loss, outperforming the vanilla version, which is consistent with the quantitative metrics, which provide stronger ROI reconstruction fidelity.

The performance gain primarily comes from smoothing the NEO signal to form a stable energy distribution, which consistently emphasizes the spike region during optimization. Different distribution variants (Laplace, Gaussian, Cauchy) provide alternative weighting shapes, but the decisive factor is the smoothed energy itself rather than the exact functional form. As a result, all EAL variants yield comparable improvements in ROI metrics, with minor differences attributable to the smoothness of the weighting curve. The final setting for our model is three window segmentation with boundaries learnable (wlr), trained by QAT with loss of EAL-G.

### 4.3 COMPARISON

For a fair comparison, we applied all the NN-based models' decoders to our designed architecture. From Table 3 and the downstream clustering results (Table 4), we can get three consistent findings.

First, large unconstrained methods (e.g., PCA, AE_FP32) attain the best absolute reconstruction metrics (highest PSNR / SNDR and lowest NRMSE) but require one to two orders of magnitude more encoder storage. Such solutions are infeasible for on-chip deployment. Conversely, extremely low-bit or binary nets (AE_INT1.4, BNN AE) reduce parameter count but suffer large drops in reconstruction fidelity and downstream clustering ability, showing the essential spike information lost under aggressive quantization.

Second, our NSC produces a practical trade-off. With only 124 bits of encoder state, NSC preserves spike-region information better than other compact alternatives (e.g., AE_INT1.4, BNN AE). Furthermore, NSC achieves ROI PSNR and ROI clustering scores close to much larger models while using $\sim 99\%$ fewer encoder bits than AE_FP32 and $\sim 97\%$ fewer than AE_QINT8. This demonstrates that the NSC design and the EAL training objective concentrate representational capacity on task-relevant (spike) regions rather than on global waveform fidelity — an intentional trade-off for implantable front-ends where spike recovery and downstream sorting matter most.

Third, the effect of EAL is interpretable and dataset-dependent. EAL-Gaussian (NSC EAL-G) consistently increases ROI metrics and downstream ROI clustering to high scores at the cost of somewhat lower FULL-PSNR and larger variance in some full-window metrics. This pattern indicates EAL shifts model capacity to spike peaks, reconstructed spikes (the biologically important region) become more accurate, while low-energy background is deprioritized.

In summary, the tables show a clear trade-off: if the requirement is strict on-chip budget with preserved spike fidelity and downstream sorting utility, NSC (with EAL when ROI fidelity is critical) offers the best practical balance. If absolute end-to-end waveform fidelity is the single objective and on-chip resources are abundant, PCA/AE_FP32 remains superior but impractical for implantable hardware. We emphasize that the NSC design intentionally sacrifices some global metrics to maximize biologically relevant reconstruction under extreme resource constraints.

### 4.4 GENERALIZATION

We've also made a small test on the MIT-BIH ECG datasets. For a fair comparison, we apply our designed decoder architecture to both the AE and our NSC model. The results (Table 5) reveal a consistent trade-off. The AE model achieves higher scores on full-segment metrics, demonstrating

Table 3: Comparison of Different Compression Models

| Model(Datasets) | $PSNR_{FULL/ROI}$ ↑ | $SNDR_{FULL/ROI}$ ↑ | $NRMSE_{FULL/ROI}$ ↓ | Encoder Params |
|---|---|---|---|---|
| AE_FP32 (GC) | 23.66 ± 1.21 / 25.97 ± 2.85 | 8.38 ± 1.21 / 13.50 ± 2.83 | 0.08 ± 0.01 / 0.06 ± 0.02 | 16512 bits |
| AE_INT8 (GC) | 22.23 ± 1.32 / 22.29 ± 1.93 | 6.94 ± 1.32 / 9.82 ± 1.92 | 0.09 ± 0.01 / 0.09 ± 0.02 | 4128 bits |
| AE_INT1.4 (GC) | 18.96 ± 2.00 / 17.95 ± 3.35 | 3.67 ± 1.98 / 5.48 ± 3.34 | 0.13 ± 0.02 / 0.16 ± 0.05 | 2580 bits |
| PCA (GC) | 29.80 ± 0.07 / 30.22 ± 0.06 | 14.51 ± 0.06 / 17.75 ± 0.06 | 0.04 ± 0.00 / 0.03 ± 0.00 | 20480 bits |
| CS (GC) | 15.47 ± 0.13 / 12.73 ± 0.20 | 0.18 ± 0.14 / 0.26 ± 0.21 | 0.17 ± 0.00 / 0.23 ± 0.01 | 16384 bits |
| VQ VAE (GC) | 18.63 ± 1.59 / 21.76 ± 4.06 | 3.34 ± 1.60 / 9.29 ± 4.07 | 0.13 ± 0.02 / 0.10 ± 0.05 | 18560 bits |
| BNN AE (GC) | 17.60 ± 1.22 / 17.19 ± 1.43 | 2.31 ± 1.23 / 4.72 ± 1.41 | 0.14 ± 0.02 / 0.15 ± 0.02 | 516 bits |
| NSC MSE (GC)* | 22.87 ± 2.43 / 23.01 ± 1.93 | 7.58 ± 2.42 / 10.54 ± 1.94 | 0.09 ± 0.02 / 0.10 ± 0.03 | 124 bits |
| NSC EAL-G (GC)* | 21.12 ± 6.38 / 25.70 ± 0.21 | 5.83 ± 6.36 / 13.23 ± 0.21 | 0.19 ± 0.16 / 0.06 ± 0.00 | 124 bits |
| AE_FP32 (hc1) | 19.44 ± 0.66 / 19.89 ± 0.84 | 3.38 ± 0.67 / 5.96 ± 0.85 | 0.11 ± 0.01 / 0.11 ± 0.01 | 16512 bits |
| AE_INT8 (hc1) | 18.47 ± 0.57 / 17.81 ± 0.86 | 2.41 ± 0.57 / 3.88 ± 0.87 | 0.13 ± 0.01 / 0.14 ± 0.01 | 4128 bits |
| AE_INT1.4 (hc1) | 16.96 ± 0.61 / 17.61 ± 1.00 | 0.90 ± 0.62 / 3.68 ± 1.01 | 0.15 ± 0.01 / 0.15 ± 0.02 | 2580 bits |
| PCA (hc1) | 21.03 ± 0.01 / 21.39 ± 0.06 | 4.98 ± 0.01 / 7.45 ± 0.04 | 0.09 ± 0.00 / 0.09 ± 0.00 | 20480 bits |
| CS (hc1) | 16.22 ± 0.07 / 14.18 ± 0.13 | 0.17 ± 0.07 / 0.25 ± 0.14 | 0.16 ± 0.00 / 0.20 ± 0.00 | 16384 bits |
| VQ VAE (hc1) | 17.29 ± 0.79 / 18.81 ± 0.81 | 1.23 ± 0.80 / 4.88 ± 0.81 | 0.15 ± 0.01 / 0.12 ± 0.01 | 18560 bits |
| BNN AE (hc1) | 16.83 ± 1.08 / 16.55 ± 1.40 | 0.77 ± 1.09 / 2.61 ± 1.41 | 0.16 ± 0.02 / 0.16 ± 0.02 | 516 bits |
| NSC MSE (hc1)* | 17.63 ± 0.52 / 18.27 ± 1.49 | 1.57 ± 0.53 / 4.34 ± 1.49 | 0.14 ± 0.01 / 0.13 ± 0.02 | 124 bits |
| NSC EAL-G (hc1)* | 17.92 ± 1.47 / 19.67 ± 0.06 | 1.86 ± 1.48 / 5.74 ± 0.05 | 0.15 ± 0.06 / 0.11 ± 0.00 | 124 bits |

· Mean ± std over five seeds.

Table 4: Clustering Results

Latent Value

| Datasets | Metrics | AE_FP32 | AE_INT8 | AE_INT1.4 | PCA | CS | VQ VAE | BNN AE | NSC MSE | NSC EAL-G |
|---|---|---|---|---|---|---|---|---|---|---|
| GC | F1 ↑ | 0.98 ± 0.01 | 0.72 ± 0.09 | 0.59 ± 0.12 | 0.88 ± 0.15 | 0.87 ± 0.11 | 0.38 ± 0.03 | 0.33 ± 0.01 | 0.63 ± 0.03 | 0.72 ± 0.12 |
| | ARI ↑ | 0.94 ± 0.04 | 0.42 ± 0.15 | 0.28 ± 0.20 | 0.83 ± 0.21 | 0.74 ± 0.18 | -0.00 ± 0.00 | 0.00 ± 0.00 | 0.33 ± 0.07 | 0.46 ± 0.20 |
| | NMI ↑ | 0.91 ± 0.05 | 0.46 ± 0.13 | 0.25 ± 0.15 | 0.87 ± 0.15 | 0.74 ± 0.15 | 0.00 ± 0.00 | 0.00 ± 0.00 | 0.35 ± 0.07 | 0.52 ± 0.20 |
| hc1 | F1 ↑ | 0.93 ± 0.02 | 0.74 ± 0.07 | 0.71 ± 0.12 | 0.97 ± 0.00 | 0.85 ± 0.06 | 0.53 ± 0.01 | 0.86 ± 0.02 | 0.68 ± 0.11 | 0.86 ± 0.12 |
| | ARI ↑ | 0.73 ± 0.06 | 0.24 ± 0.13 | 0.23 ± 0.22 | 0.87 ± 0.02 | 0.49 ± 0.17 | 0.00 ± 0.00 | 0.54 ± 0.04 | 0.16 ± 0.10 | 0.58 ± 0.31 |
| | NMI ↑ | 0.69 ± 0.06 | 0.22 ± 0.16 | 0.18 ± 0.18 | 0.81 ± 0.02 | 0.47 ± 0.14 | 0.00 ± 0.00 | 0.49 ± 0.03 | 0.30 ± 0.08 | 0.59 ± 0.22 |

· Mean ± std over five seeds.

Reconstructed Waveform Full

| Datasets | Metrics | AE_FP32 | AE_INT8 | AE_INT1.4 | PCA | CS | VQ VAE | BNN AE | NSC MSE | NSC EAL-G |
|---|---|---|---|---|---|---|---|---|---|---|
| GC | F1 ↑ | 0.88 ± 0.14 | 0.69 ± 0.06 | 0.60 ± 0.11 | 0.88 ± 0.15 | 0.87 ± 0.11 | 0.79 ± 0.24 | 0.34 ± 0.01 | 0.80 ± 0.10 | 0.68 ± 0.19 |
| | ARI ↑ | 0.82 ± 0.19 | 0.47 ± 0.07 | 0.28 ± 0.19 | 0.83 ± 0.21 | 0.74 ± 0.18 | 0.64 ± 0.35 | -0.00 ± 0.00 | 0.66 ± 0.16 | 0.49 ± 0.26 |
| | NMI ↑ | 0.85 ± 0.13 | 0.49 ± 0.08 | 0.27 ± 0.16 | 0.87 ± 0.15 | 0.74 ± 0.15 | 0.60 ± 0.33 | 0.00 ± 0.00 | 0.71 ± 0.13 | 0.56 ± 0.26 |
| hc1 | F1 ↑ | 0.89 ± 0.06 | 0.68 ± 0.16 | 0.75 ± 0.05 | 0.97 ± 0.00 | 0.85 ± 0.06 | 0.96 ± 0.01 | 0.94 ± 0.00 | 0.95 ± 0.01 | 0.64 ± 0.21 |
| | ARI ↑ | 0.61 ± 0.18 | 0.23 ± 0.21 | 0.26 ± 0.09 | 0.87 ± 0.02 | 0.49 ± 0.17 | 0.85 ± 0.05 | 0.77 ± 0.01 | 0.81 ± 0.02 | 0.24 ± 0.35 |
| | NMI ↑ | 0.61 ± 0.13 | 0.22 ± 0.20 | 0.21 ± 0.10 | 0.81 ± 0.02 | 0.47 ± 0.14 | 0.78 ± 0.06 | 0.66 ± 0.01 | 0.74 ± 0.02 | 0.32 ± 0.27 |

· Mean ± std over five seeds.

Reconstructed Waveform ROI

| Datasets | Metrics | AE_FP32 | AE_INT8 | AE_INT1.4 | PCA | CS | VQ VAE | BNN AE | NSC MSE | NSC EAL-G |
|---|---|---|---|---|---|---|---|---|---|---|
| GC | F1 ↑ | 0.94 ± 0.12 | 0.75 ± 0.11 | 0.62 ± 0.13 | 0.88 ± 0.15 | 0.89 ± 0.07 | 0.79 ± 0.24 | 0.34 ± 0.01 | 0.80 ± 0.10 | 1.00 ± 0.00 |
| | ARI ↑ | 0.90 ± 0.16 | 0.54 ± 0.13 | 0.31 ± 0.21 | 0.83 ± 0.21 | 0.72 ± 0.18 | 0.64 ± 0.35 | -0.00 ± 0.00 | 0.66 ± 0.16 | 0.99 ± 0.01 |
| | NMI ↑ | 0.91 ± 0.12 | 0.55 ± 0.12 | 0.29 ± 0.17 | 0.87 ± 0.15 | 0.73 ± 0.15 | 0.60 ± 0.33 | 0.00 ± 0.00 | 0.71 ± 0.13 | 0.98 ± 0.01 |
| hc1 | F1 ↑ | 0.94 ± 0.03 | 0.83 ± 0.06 | 0.79 ± 0.07 | 0.97 ± 0.00 | 0.82 ± 0.08 | 0.96 ± 0.01 | 0.94 ± 0.00 | 0.95 ± 0.01 | 0.96 ± 0.01 |
| | ARI ↑ | 0.76 ± 0.11 | 0.44 ± 0.15 | 0.35 ± 0.16 | 0.87 ± 0.01 | 0.44 ± 0.20 | 0.86 ± 0.05 | 0.77 ± 0.01 | 0.81 ± 0.02 | 0.85 ± 0.02 |
| | NMI ↑ | 0.72 ± 0.09 | 0.38 ± 0.13 | 0.28 ± 0.14 | 0.81 ± 0.02 | 0.43 ± 0.18 | 0.78 ± 0.05 | 0.66 ± 0.01 | 0.74 ± 0.02 | 0.78 ± 0.02 |

· Mean ± std over five seeds.

its capacity for global signal reconstruction. In contrast, our NSC model sacrifices some global fidelity but consistently and significantly outperforms the AE in reconstructing the Regions of Interest (ROI). These results confirm that the NSC framework generalizes effectively beyond neural signals and maintains excellent, focused reconstruction quality for critical waveform segments in other biosignals like ECG.

Table 5: Generalization on MIT-BIH datasets

| Datasets | Model | $PSNR_{FULL/ROI}$ ↑ | $SNDR_{FULL/ROI}$ ↑ | $NRMSE_{FULL/ROI}$ ↓ |
|---|---|---|---|---|
| 100 | AE_FP32 | **20.70 ± 1.49** / 18.90 ± 2.89 | **8.83 ± 1.48** / 8.10 ± 2.87 | **0.09 ± 0.02** / 0.12 ± 0.03 |
| | NSC EAL-G | 17.73 ± 5.28 / **28.96 ± 0.38** | 5.86 ± 5.30 / **18.16 ± 0.37** | 0.16 ± 0.09 / **0.04 ± 0.00** |
| 101 | AE_FP32 | **21.65 ± 2.36** / 19.15 ± 2.35 | **9.01 ± 2.37** / 7.78 ± 2.36 | **0.09 ± 0.02** / 0.12 ± 0.02 |
| | NSC EAL-G | 17.51 ± 2.91 / **29.06 ± 0.80** | 4.87 ± 2.89 / **17.69 ± 0.79** | 0.16 ± 0.04 / **0.05 ± 0.01** |
| 102 | AE_FP32 | 19.47 ± 2.29 / 18.81 ± 5.08 | 7.42 ± 2.29 / 6.43 ± 5.09 | **0.13 ± 0.02** / 0.14 ± 0.05 |
| | NSC EAL-G | **21.78 ± 6.86 / 27.59 ± 0.33** | **9.72 ± 6.86 / 15.21 ± 0.31** | 0.16 ± 0.13 / **0.05 ± 0.00** |

· Mean ± std over five seeds.

## 5 CONCLUSION

This paper presented a Neural Signal Codec (NSC) featuring a highly resource-efficient encoder for implantable brain-machine interface systems. The proposed NSC employs a hardware-optimized encoder with only 124 bits of parameters and is trained with an Energy-Aware Loss (EAL), 97% parameter reduction compared with conventional AE_QINT8, achieving an average PSNR of more than 17.92 dB at a 32:1 compression ratio across all datasets. High-fidelity reconstructions within the region of interest and downstream clustering experiments show that the proposed NSC excels at preserving spike-related information compared to conventional parameter-intensive models.

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

# Appendix

## A NSC ENCODER ANALYSIS

### A.1 BACKWARD DERIVATIONS

Although designed encoder applies parameter quantization during the forward pass to emulate low-bit hardware behavior, the backward pass computes gradients using the full-precision parameters. Gradients are computed with respect to the input signal $\mathbf{x}$, the energy-aware parameters $\boldsymbol{\alpha}, \boldsymbol{\beta}, \boldsymbol{\gamma}$, and the window boundary parameters $\boldsymbol{r}$, all treated as continuous variables during learning. And let $\nabla_z \mathcal{L}$ be the gradient acquired form decoder.

**Gradient w.r.t. Input x.** For each window $i$, the local gradient with respect to each time point $x_n^{(i)}$ is:

$$\frac{\partial \mathcal{L}}{\partial x_n^{(i)}} = \nabla_z \mathcal{L} \cdot \gamma_i \cdot \sum_{j=1}^{d} \text{Scale}_{j,n}^{(i)}. \tag{10}$$

These window-local gradients are then aggregated into the corresponding region of the full-length gradient $\nabla_{\mathbf{x}}$.

**Gradient w.r.t. $\alpha_{i,j}$ and $\beta_{i,j}$.** With the chain rule, we obtain:

$$\frac{\partial \mathcal{L}}{\partial \alpha_{i,j}} = \nabla_z \mathcal{L} \cdot \gamma_i \cdot \ln(2) \cdot \sum_{n=1}^{L_i} x_n^{(i)} \cdot \text{LNE}_n^{(i)} \cdot \text{Scale}_{j,n}^{(i)}. \tag{11}$$

$$\frac{\partial \mathcal{L}}{\partial \beta_{i,j}} = \nabla_z \mathcal{L} \cdot \gamma_i \cdot \ln(2) \cdot \sum_{n=1}^{L_i} x_n^{(i)} \cdot \text{Scale}_{j,n}^{(i)}. \tag{12}$$

**Gradient w.r.t.** $\gamma_i$. Since $\gamma_i$ acts as a linear scaling factor on the window output, the gradient is computed using the unscaled window output (fake y) for numerical stability and mathematical correctness:

$$\frac{\partial \mathcal{L}}{\partial \gamma_i} = \nabla_z \mathcal{L} \cdot y_{fake}^{(i)} \tag{13}$$

where $y_{fake}^{(i)}$ is the window output with $\gamma_i = 1.0$.

**Gradient w.r.t Quantized Parameters.** During training we apply the straight-through estimator (STE) Bengio et al. (2013) for parameters and floor operations to enable gradient propagation to achieve the fake quant with:

$$(\theta_{quant} - \theta).detach() + \theta \tag{14}$$

Specifically, for the quantized parameter $\theta_q$, we approximate:

$$\frac{\partial \theta_q}{\partial \theta} \approx 1. \tag{15}$$

allowing gradients with respect to $\theta_q$ to be directly propagated and used to update $\theta$. The gradients of the quantized parameters $\alpha_{q,i,j}, \beta_{q,i,j}, \gamma_{q,i,j}$ are back-propagated to their full-precision parts $\alpha_{i,j}, \beta_{i,j}, \gamma_{i,j}$ without modification.

**Gradient w.r.t. Window Boundary Parameters** $p$. Let the scalar boundary parameters be $p_i$ for $i = 0, \ldots, w - 1$, where $w$ is the number of windows. Define $s_i = \sigma(p_i) = \text{sigmoid}(p_i)$, optionally clamped in implementation for stability.

The normalized boundary positions $b_i \in [0, 1]$ are defined recursively as:

$$b_0 = 0, \quad b_i = 1 - \prod_{t=0}^{i-1}(1 - s_t), \quad i = 1, \ldots, w \tag{16}$$

which is equivalent to the code's recurrence: $b_i = b_{i-1} + (1 - b_{i-1})s_{i-1}$. The physical boundary positions for input length $T$ are:

$$B_i = T \cdot b_i, \quad i = 0, \ldots, w \tag{17}$$

Each window $i$ produces an unscaled output $y(i)_{fake}$ (data on window $[B_{i-1}, B_i]$). The gradient contribution for window $i$ is:

$$G_i = \sum_{b \in \text{batch}} \left( \nabla_z \mathcal{L}_b \cdot y_{fake}^{(i)} \right) \tag{18}$$

The gradient with respect to boundary parameter $p_i$ is derived as follows:

1) BOUNDARY PERTURBATION EFFECT: A small perturbation $\delta$ in boundary $B_{i+1}$ redistributes samples between windows $i$ and $i + 1$:

$$\frac{\partial \mathcal{L}}{\partial B_{i+1}} = G_{i+1} - G_i \tag{19}$$

2) WITH CHAIN RULE:

$$\frac{\partial \mathcal{L}}{\partial p_i} = \frac{\partial \mathcal{L}}{\partial B_{i+1}} \cdot \frac{\partial B_{i+1}}{\partial p_i} \tag{20}$$

3) BOUNDARY POSITION DERIVATIVE: Differentiating $b_k = 1 - \prod_{t=0}^{k-1}(1 - s_t)$ with respect to $s_i$ (for $i < k$):

$$\frac{\partial b_k}{\partial s_i} = \prod_{\substack{t=0 \\ t \neq i}}^{k-1}(1 - s_t) \tag{21}$$

For $k = i + 1$:

$$\frac{\partial b_{i+1}}{\partial s_i} = \prod_{t=0}^{i-1}(1 - s_t) \tag{22}$$

4) SIGMOID DERIVATIVE:

$$\frac{\partial s_i}{\partial p_i} = s_i(1 - s_i) \tag{23}$$

5) COMBINED DERIVATIVE:

$$\frac{\partial B_{i+1}}{\partial p_i} = T \cdot \frac{\partial b_{i+1}}{\partial s_i} \cdot \frac{\partial s_i}{\partial p_i} = T \cdot \left(\prod_{t=0}^{i-1}(1 - s_t)\right) \cdot s_i(1 - s_i) \tag{24}$$

6) FINAL GRADIENT EXPRESSION:    Combining equations (19) and (24):

$$\frac{\partial \mathcal{L}}{\partial p_i} = (G_{i+1} - G_i) \cdot T \cdot \left(\prod_{t=0}^{i-1}(1 - s_t)\right) \cdot s_i(1 - s_i) \tag{25}$$

NOTED:

- The code implements this exactly: `window_aggr` stores $G_k$ values, `ds` $= s_i(1 - s_i)$, and `d_boundaries` $= T \cdot \prod_{t=0}^{i-1}(1 - s_t) \cdot \text{ds}[i]$.
- Intuition: $(G_{i+1} - G_i)$ measures whether moving the boundary helps loss reduction.

## A.2    BOUNDS AND GRADIENTS

In this section, we mainly study the bounds for encoder parameters, variables, and outputs.

### A.2.1    DISTRIBUTION OF NEO ($T = X^2 - YZ$)

**Discrete Input Analysis**   : Let $X, Y, Z$ be independent and uniformly distributed over the 8-bits signed integer set $\{-128, \ldots, 127\} \cup \{128\}$ for convenience, each with

$$P(X = x) = P(Y = y) = P(Z = z) = \frac{1}{257}, \quad x, y, z \in \{-128, -127, \ldots, 128\}.$$

Define T(NEO) with :
$$T = X^2 - YZ.$$

then :
$$X^2 \in \{0^2, 1^2, \ldots, 128^2\}, \quad YZ \in \{-128 \cdot 128, -127 \cdot 128, \ldots, 128 \cdot 128\},$$
and consequently :
$$T \in \{-128^2, \ldots, 2 \cdot 128^2\}.$$

Then the probability mass function of $T$ is given by :

$$P(T = t) = \frac{N(t)}{257^3}, \quad \text{where} \quad N(t) = \left|\{(x, y, z) \in \{-128, \ldots, 128\}^3 : x^2 - yz = t\}\right|.$$

Mean:
$$\mathbb{E}[T] = \mathbb{E}[X^2] - \mathbb{E}[Y]\,\mathbb{E}[Z] = \mathbb{E}[X^2],$$
since $\mathbb{E}[Y] = \mathbb{E}[Z] = 0$. Moreover :

$$\mathbb{E}[T] = \mathbb{E}[X^2] = \frac{1}{257} \sum_{k=-128}^{128} k^2 = \frac{1}{257} \cdot 2 \cdot \frac{128 \cdot (128 + 1) \cdot (2 \cdot 128 + 1)}{6} = 5504.$$

Variance:
$$\text{Var}(T) = \text{Var}(X^2) + \text{Var}(YZ),$$
where $X, Y, Z$ are mutually independent. Since $Y, Z$ are independent and symmetric :

$$\text{Var}(YZ) = \mathbb{E}[Y^2]\,\mathbb{E}[Z^2], \quad \text{and} \quad \mathbb{E}[Y^2] = \mathbb{E}[Z^2] = \frac{1}{257} \sum_{k=-128}^{128} k^2 = 5504.$$

Next, for $\mathrm{Var}(X^2)$:

$$\mathrm{Var}(X^2) = \mathbb{E}[X^4] - \left(\mathbb{E}[X^2]\right)^2.$$

We have

$$\mathbb{E}[X^4] = \frac{1}{257} \sum_{k=-128}^{128} k^4 = \frac{1}{257} \cdot 2 \cdot \sum_{k=1}^{128} k^4 = \frac{2}{257} \cdot \frac{128(128+1)(2 \cdot 128 + 1)(3 \cdot 128^2 + 3 \cdot 128 - 1)}{30}.$$

This simplifies to:

$$\mathbb{E}[X^4] = \frac{2}{257} \cdot \frac{128 \cdot 129 \cdot 257 \cdot 49535}{30} = \frac{2 \cdot 128 \cdot 129 \cdot 49535}{30}.$$

So finally, the total variance is

$$\mathrm{Var}(T) = \left(\mathbb{E}[X^4] - \mathbb{E}[X^2]^2\right) + \mathrm{Var}(YZ) = \left(\frac{2 \cdot 128 \cdot 129 \cdot 49535}{30} - 5504^2\right) + 5504^2 \approx 5.453 \times 10^7.$$

**Continuous Input Analysis** : $X, Y, Z \in \mathbb{R} \sim \mathcal{U}(-a, a)$ for $a = 128$. Then, the PDF of $T = X^2 - YZ$ becomes:

$$f_T(t) = \int_0^{a^2} f_{X^2}(s) \cdot f_{YZ}(s-t) \, ds$$

Where:

$$f_{X^2}(s) = \begin{cases} \dfrac{1}{2a\sqrt{s}}, & 0 < s \le a^2 \\[2mm] 0, & \text{otherwise} \end{cases}$$

$$f_{YZ}(b) = \begin{cases} \dfrac{1}{2a^2}, & b = 0 \\[2mm] -\dfrac{1}{2a^2} \log\left(\dfrac{|b|}{a^2}\right), & 0 < |b| < a^2 \\[2mm] 0, & \text{otherwise} \end{cases}$$

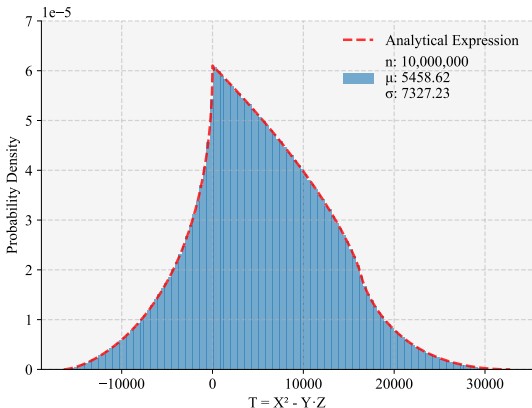

Figure 4: Discrete Empirical Distribution of $T = X^2 - YZ$ (histogram) and corresponding Continuous Analytical Expression (red curve)

Code experiment results are shown in Fig.4.

### A.2.2    DISTRIBUTION OF LNE ($\lfloor \log_2(\max(T, 1.0)) \rfloor$)

We define the discrete LNE as:

$$\text{LNE} = \lfloor \log_2\left(\max(T, 1.0)\right) \rfloor.$$

**Discrete Input Analysis**    For values $T < 1$, LNE is set to 0. The probability mass function is computed as:

$$P(\text{LNE} = k) = \begin{cases} \dfrac{1}{257^3} \displaystyle\sum_{t=-128^2}^{1} N(t), & k = 0, \\[2em] \dfrac{1}{257^3} \displaystyle\sum_{t=2^k}^{2^{k+1}-1} N(t), & k \geq 1. \end{cases}$$

**Continuous Input Analysis**    For the continuous analytical expression, floor was ignored and let $y = \log_2(\max(T, 1))$, then $T = 2^y$. The probability density transforms as:

$$f_Y(y) = f_T(2^y) \cdot \left| \frac{dT}{dy} \right| = f_T(2^y) \cdot 2^y \log(2) \quad y > 0, \quad else \quad f_Y(y) = \int_{-\infty}^{1} f_T(t)dt \quad y = 0$$

The distribution of LNE is obtained by integrating over bins:

$$P(\text{LNE} = k) \approx \begin{cases} \displaystyle\int_{-\infty}^{1} f_Y(y)dy, & k = 0, \\[1.5em] \displaystyle\int_{k}^{k+1} f_Y(y)dy, & k \geq 1. \end{cases}$$

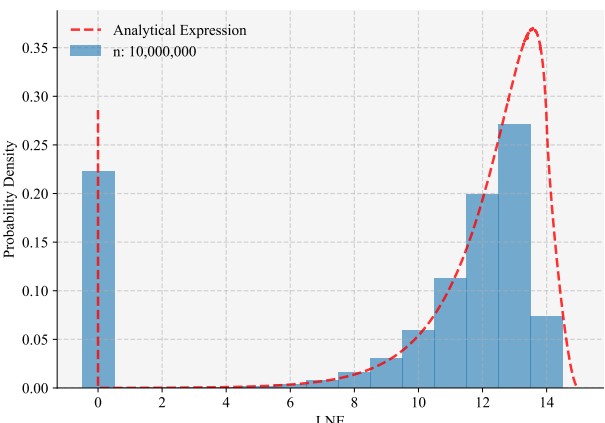

Figure 5: Discrete Empirical Distribution of LNE (histogram) and corresponding Continuous Analytical Expression (red curve)

Code experiment results are shown in Fig.5.

### A.2.3    DISTRIBUTION OF SCALE VALUE

In forward processing part, for window $i$ we have:

$$\text{Shift}_{j,n}^{(i)} = \lfloor \alpha_{q,i,j} \cdot \text{LNE}_n + \beta_{q,i,j} \rfloor,$$

$$\text{Scale}_{j,n}^{(i)} = 2^{\text{clamp}(\text{Shift}_{j,n}^{(i)} - 8, -8, 7)}.$$

Where $\alpha_{q,i,j} \in \{0, 1/16, ...15/16\}$ and $\beta_{q,i,j} \in \{-8, -7, ...7\}$, here, we consider the continuous $\tilde{\alpha} \in \mathbb{R} \sim \mathcal{U}(0, 1)$ and $\tilde{\beta} \in \mathbb{Z} \sim \mathcal{U}(-8, 8)$, and let X $\sim P(\text{LNE})$. The distribution for LNE is

presented in A.2.2, results show that LNE has bimodal characteristics, one located in $0$ and another is around value of $\sim 13$. Considering the uniform distribution of $\tilde{\alpha}$ and $\tilde{\beta}$, the range of Shift $\in (-8, 24)$, here we choose bias of $-8$ as default to change the Shift range to the symmetrical interval in $(-16, 16)$.

Let's consider :

$$Z = Y + \tilde{\beta}, \quad Y = \tilde{\alpha} \cdot X$$

and mainly discuss of the scale value of :

$$scale\, value = clamp(Z - 8, -8, 7)$$

and $\tilde{\alpha}$, $\tilde{\beta}$ are independent of each other and of $X$. First considering $Y$, we can get the cumulative distribution function (CDF) :

$$F_Y(y) = P(Y \le y) = P(\tilde{\alpha} \cdot X \le y)$$

$$= \int_0^\infty P(\tilde{\alpha} \le \frac{y}{x} \mid x = X) \cdot f_X(x) dx$$

$$= \int_0^y 1 \cdot f_X(x) dx + \int_y^{16} \frac{y}{x} f_X(x) dx \quad (\tilde{\alpha} \sim \mathcal{U}(0, 1))$$

$$= F_X(y) + y \int_y^{16} \frac{f_X(x)}{x} dx$$

and the probability density function (PDF) with:

$$f_Y(y) = \frac{d}{dy} F_Y(y) = f_X(y) + \int_y^{16} \frac{f_X(x)}{x} dx + y \cdot (-\frac{f_X(y)}{y}) = \int_y^{16} \frac{f_X(x)}{x} dx$$

Then the PDF of $Z$ is given by the convolution:

$$f_Z(z) = \int_{-\infty}^\infty f_Y(z - b) f_{\tilde{\beta}}(b) \, db$$

Since $\tilde{\beta} \sim \mathcal{U}(-8, 8)$, we have $f_{\tilde{\beta}}(b) = \frac{1}{16}$ for $b \in [-8, 8]$, so:

$$f_Z(z) = \frac{1}{16} \int_{-8}^8 f_Y(z - b) \, db = \frac{1}{16} \int_{-8}^8 \left( \int_{z-b}^{16} \frac{f_X(x)}{x} \, dx \right) db$$

And for the statistical values, we can get:

Expected Value:

$$\mathbb{E}[Z] = \mathbb{E}[Y] + \mathbb{E}[\tilde{\beta}] = \mathbb{E}[\tilde{\alpha}] \cdot \mathbb{E}[X] + 0 = \frac{1}{2} \mathbb{E}[X]$$

Variance:

$$\mathrm{Var}(Z) = \mathrm{Var}(Y) + \mathrm{Var}(\tilde{\beta})$$

$$\mathrm{Var}(\tilde{\beta}) = \frac{(8 - (-8))^2}{12} = \frac{256}{12} = \frac{64}{3}$$

$$\mathrm{Var}(Y) = \mathbb{E}[\mathrm{Var}(Y \mid X)] + \mathrm{Var}(\mathbb{E}[Y \mid X]) = \frac{1}{12} \mathbb{E}[X^2] + \frac{1}{4} \mathrm{Var}(X)$$

In A.2.2 we can get $\mathbb{E}[X(LNE)] \approx 12$, $\mathrm{Var}[X] \approx 2.6$ (ignoring $0$ value, for it makes no contributions for $\alpha, \beta$ gradient, and as mentioned the low NEO represents the region more like noise), hence the final :

$$\mathbb{E}[Z] = \frac{1}{2} \mathbb{E}[X] \approx 5$$

$$\mathrm{Var}(Z) = \frac{1}{12} \mathbb{E}[X^2] + \frac{1}{4} \mathrm{Var}(X) + \frac{64}{3} \approx 37 \approx 6^2$$

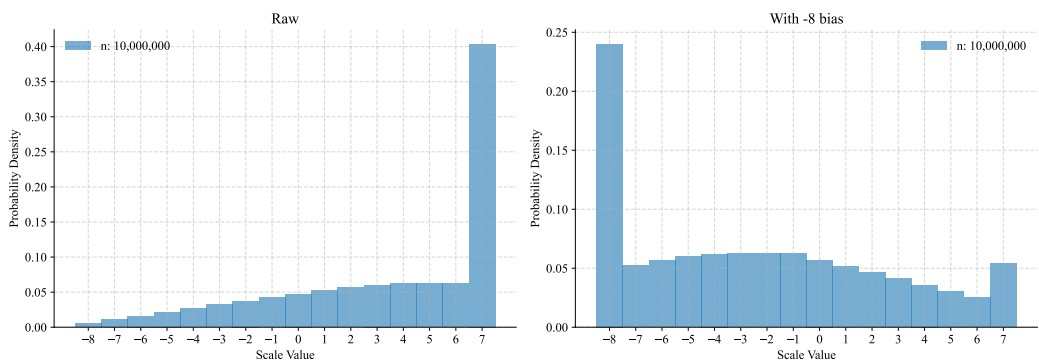

Figure 6: Discrete Empirical Distribution of Scale Value (histogram), Left: Original Scale Distribution. Right: With $-8$ Bias Scale Distribution

By adding the bias $-8$, we can make the distribution centered to $\mathbb{E}[Z-8] \approx -3$, with the distribution figure shows below :

Fig.6 shows $\mathbb{E}[scale\,value] \approx -2.46 \quad \mathrm{Var}[scale\,value] \approx 22.78 \approx 4.77^2$, which is consistent with the theoretical results. The majority of scale values are concentrated on the negative axis, which a property for numerical stability. Since computation involves scaling by $x \cdot 2^{\mathrm{scale}}$. If scale were predominantly positive, the output could grow explosively (as illustrated in the left figure).

### A.2.4 GRADIENT BOUND ANALYSIS

For $\nabla_z \mathcal{L}$ is given through the decoder, usually considered to be stable:

$$|\nabla_z \mathcal{L}| \leq C$$

For the upper bound of extreme value, we choose all the possible max value of $X = 128, \mathrm{LNE} = 16, \mathrm{Scale} = 2^7$ treating all to be independent, with all windows collapsed into one of $\mathrm{T} = 128$ and $\gamma = 1$, we get:

$$\left|\frac{\partial \mathcal{L}}{\partial \alpha_{i,j}}\right| \leq C \cdot 1 \cdot \ln(2) \cdot 128 \cdot (128 \cdot 16 \cdot 128)$$
$$\approx C \cdot 2.3 \times 10^7$$
$$\left|\frac{\partial \mathcal{L}}{\partial \beta_{i,j}}\right| \leq C \cdot 1 \cdot \ln(2) \cdot 128 \cdot (128 \cdot 128)$$
$$\approx C \cdot 1.5 \times 10^6$$

However, this bound considers the joint distribution of input signal, NEO, and scale. Here we adopt a Monte Carlo approach:

**Parameter Distributions**:

- N times testing, with window length $L$ of 128.
- Setting $\mathrm{X} \in \mathbb{Z}^{N \times 130} \sim \mathcal{U}\{-128, 127, ..., 127\}$, with padding 2 to get $128$ length NEO value.
- $\alpha, \gamma \in \mathbb{R}^N \sim \mathcal{U}(0, 15/16), \beta \in \mathbb{Z}^N \sim \mathcal{U}(-8, 7)$, considering the STE in training process.
- For the window boundaries, we choose $s_0, s_1 \in \mathbb{R}^N \sim \mathcal{U}(0.05, 0.95)$ for stability, with boundaries $b_0 = 0, b_1 = s_0, b_2 = b_1 + (1 - b_1) \cdot s_1, b_3 = 1$. The actual boundary position is $B_k = b_k \cdot L$.

Figure 7 shows the detailed distribution of gradient $\|\alpha\|$ and $\|\beta\|$.

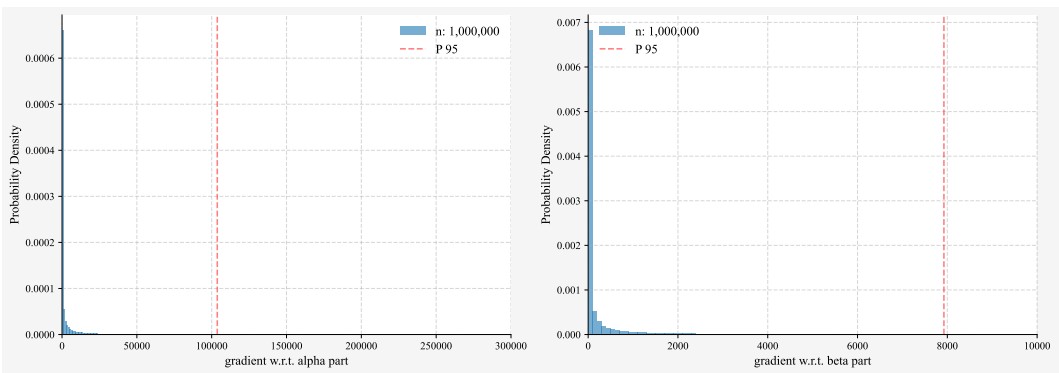

Figure 7: Gradient distribution w.r.t. $\|\alpha\|$ and $\|\beta\|$

And we get the tight upper bound of experience with:

$$\left|\frac{\partial \mathcal{L}}{\partial \alpha_{i,j}}\right| \approx C \cdot 2.06 \times 10^6, \quad \left|\frac{\partial \mathcal{L}}{\partial \beta_{i,j}}\right| \approx C \cdot 1.04 \times 10^5$$

$$\left.\left|\frac{\partial \mathcal{L}}{\partial \alpha_{i,j}}\right|\right|_{P_{95}} \approx C \cdot 1.64 \times 10^5, \quad \left.\left|\frac{\partial \mathcal{L}}{\partial \beta_{i,j}}\right|\right|_{P_{95}} \approx C \cdot 7.92 \times 10^3$$

$$P(\left|\frac{\partial \mathcal{L}}{\partial \alpha_{i,j}}\right| == 0) \approx 0.39\%, P(\left|\frac{\partial \mathcal{L}}{\partial \beta_{i,j}}\right| == 0) \approx 0.15\%$$

While the gradient bounds are derived per-sample, the averaging nature of mini-batches further reduces the effective variance and magnitude, ensuring training stability.

### A.3 LIPSCHITZ CONTINUITY PROOF

Considering :

$$\text{Scale}_n = 2^{\alpha \cdot \log_2(\max(\text{NEO}_n, 1)) + \beta - 8} = 2^{\beta - 8} \cdot (\max(\text{NEO}_n, 1))^\alpha \leq (\text{M}_n)^\alpha, \quad \text{M}_n = \max(\text{NEO}_n, 1)$$

Define :

$$g(\alpha) = C \cdot \sum_{n=1}^{L} x_n \cdot \text{LNE}_n \cdot \text{M}_n{}^\alpha, \quad C = \nabla_z \mathcal{L} \cdot \gamma \cdot \ln(2)$$

Let :

$$\phi_n(\alpha) = x_n \cdot \text{LNE}_n \cdot \text{M}_n{}^\alpha$$

Then :

$$|g(\alpha_1) - g(\alpha_2)| = C \cdot \left|\sum_{n=1}^{L} (\phi_n(\alpha_1) - \phi_n(\alpha_2))\right| \leq C \cdot \sum_{n=1}^{L} |\phi_n(\alpha_1) - \phi_n(\alpha_2)|$$

Note $\phi_n(\alpha)$ is differentiable in $\alpha$:

$$\frac{d\phi_n}{d\alpha} = x_n \cdot \text{LNE}_n \cdot \text{M}^\alpha \cdot \ln(\text{M}_n) = x_n \cdot \text{LNE}_n \cdot \text{M}^\alpha \cdot \text{LNE}_n \cdot \ln(2) = \phi_n(\alpha) \cdot \text{LNE}_n \cdot \ln(2)$$

Thus :

$$|\phi_n(\alpha_1) - \phi_n(\alpha_2)|$$

$$\leq \sup_{\alpha \in [\alpha_1, \alpha_2]} \left| \frac{d\phi_n}{d\alpha} \right| \cdot |\alpha_1 - \alpha_2|$$

$$\Rightarrow |g(\alpha_1) - g(\alpha_2)| \leq C \cdot \sum_{n=1}^{L} \left| \frac{d\phi_n}{d\alpha} \right|_{\max} \cdot |\alpha_1 - \alpha_2|$$

$$\Rightarrow |g(\alpha_1) - g(\alpha_2)| \leq K_\alpha \cdot |\alpha_1 - \alpha_2|,$$

$$\text{where} \quad K_\alpha = C \cdot \ln(2) \cdot \sum_{n=1}^{L} |x_n \cdot \text{LNE}_n^2 \cdot \text{M}_n^{\ max(\alpha_1, \alpha_2)}|$$

Lipschitz continuity of gradient w.r.t. $\beta$ is similar as $\alpha$, with :

$$K_\beta = C \cdot \ln(2) \cdot \sum_{n=1}^{L} |x_n \cdot \text{LNE}_n \cdot \text{M}_n^{\ max(\alpha_1, \alpha_2)}|$$

The gradients with respect to the encoder parameters $\alpha$ and $\beta$ are both bounded and Lipschitz continuous. Ensuring the gradient magnitude remains within a finite range and guaranteeing that small perturbations in parameters induce only small changes in the gradients. Together, these properties imply that the encoder exhibits relatively stable gradient behavior.

# B HARDWARE IMPLEMENTATION DETAILS

## B.1 MULTIPLIER-FREE DESIGN

In this section, we mainly introduce the design for multiplications.

### B.1.1 $\lfloor log_2(max(neo, 1)) \rfloor$ OPERATION

For the $\lfloor log_2(x) \rfloor$ operation, for a binary number $x$, $\lfloor log_2(x) \rfloor$ is equivalent to finding the highest '1' bit. One traditional solution is the priority encoder (PE). The main idea is to find the highest bit '1' from top to bottom. This can also be achieved by nesting multiple layers of if to determine the highest bit.

We implement the log operation using a hierarchical priority encoder (PE) structure. The 16-bit input is divided into four 4-bit groups, each processed by a small 4-bit LOD module, and the outputs are combined to generate the 4-bit log value. The detailed implementation can be found in Algorithm 1.

### B.1.2 WEIGHT GENERATION PROCESS

In our design, two quantized decimal parameters $\alpha$ and $\gamma$, each stored as 4-bit values, are utilized. Both parameters represent fractional values in the set $\{0, \frac{1}{16}, \frac{2}{16}, \dots, \frac{15}{16}\}$. This section focuses on the computation of $\alpha \cdot LNE$, where $LNE$ is the 4-bit logarithm approximation obtained from Algorithm 1.

**Algorithm Overview** The multiplication $\alpha \cdot LNE$ is implemented using a shift-and-add approach that leverages the binary representation of $\alpha$. Given that $\alpha$ is a 4-bit fractional number, it can be expressed as:

$$\alpha = \frac{\alpha_3 \cdot 2^{-1} + \alpha_2 \cdot 2^{-2} + \alpha_1 \cdot 2^{-3} + \alpha_0 \cdot 2^{-4}}{1}$$

where $\alpha[3:0]$ are the individual bits of $\alpha$.

**Shift Operation Phase** The 4-bit $LNE$ value is first expanded to 8-bit precision and shifted according to the weight of each bit in $\alpha$:

---

**Algorithm 1:** Hierarchical Priority Encoder for Base-2 Floor Logarithm

---

**Input:** $data_i[15:0]$, 16-bits neo input
**Output:** $LNE[3:0]$, 4-bits log representation

**Step 1: Group-wise Detection**;
```
// Check which 4-bit groups contain at least one '1' bit
```
$zdet[3] \leftarrow data\_i[15] \vee data\_i[14] \vee data\_i[13] \vee data\_i[12]$
$zdet[2] \leftarrow data\_i[11] \vee data\_i[10] \vee data\_i[9] \vee data\_i[8]$
$zdet[1] \leftarrow data\_i[7] \vee data\_i[6] \vee data\_i[5] \vee data\_i[4]$
$zdet[0] \leftarrow data\_i[3] \vee data\_i[2] \vee data\_i[1] \vee data\_i[0]$
```
// Check if entire input is zero
```
$zero\_o \leftarrow \neg(zdet[3] \vee zdet[2] \vee zdet[1] \vee zdet[0])$

**Step 2: Leading One Detection (LOD)**;
```
// For each zdet, find the position of the leading '1'.  LOD
    module acquires 4-bits input with output 4-bits.  In LOD
    module, we have :
```
$[3:0]lod\_i, [3:0]lod\_o$;
$mux2 = (lod\_i[3] == 1) \quad ? \quad 0 \quad : \quad 1$;
$mux1 = (lod\_i[2] == 1) \quad ? \quad 0 \quad : \quad mux2$;
$mux0 = (lod\_i[1] == 1) \quad ? \quad 0 \quad : \quad mux1$;
$lod\_o[3] = lod\_i[3]$;
$lod\_o[2] = mux2 \quad \& \quad lod\_i[2]$;
$lod\_o[1] = mux1 \quad \& \quad lod\_i[1]$;
$lod\_o[0] = mux0 \quad \& \quad lod\_i[0]$;

**Step 3: Inter-group Priority Selection**;
```
// Determine which 4-bit group contains the globally highest
    '1' bit
```
$select[3:0] = LOD(zdet[3:0])$ `// For example, if` $select[3] = 1$`, which`
```
    means highest '1' in bits 15-12 (group 3)
```

**Step 4: Hierarchical Result Multiplexing**;
```
// Propagate only the leading one detection results from the
    selected group
```
**for** $group \leftarrow 3$ **downto** $0$ **do**
   $data\_o[15:12] \leftarrow (group == 3) ? LOD(data\_i[15:12]) : 4'b0000$;
   $data\_o[11:8] \leftarrow (group == 2) ? LOD(data\_i[11:8]) : 4'b0000$;
   $data\_o[7:4] \leftarrow (group == 1) ? LOD(data\_i[7:4]) : 4'b0000$;
   $data\_o[3:0] \leftarrow (group == 0) ? LOD(data\_i[3:0]) : 4'b0000$;
   `// Only the selected group's LOD results are preserved`
**end**

**Step 5: Position Encoding**;
**if** $zero\_o = 1$;                 `// Special case:  input bits are all 0`
 **then**
   | $LNE[3:0] \leftarrow 4'b0000$
**else**
   : `// Encode position using combinatorial OR logic, maps`
       $max(data\_i, 1)$
   $LNE[3] \leftarrow$
   $data\_o[14] \vee data\_o[13] \vee data\_o[12] \vee data\_o[11] \vee data\_o[10] \vee data\_o[9] \vee data\_o[8]$
   $LNE[2] \leftarrow$
   $data\_o[14] \vee data\_o[13] \vee data\_o[12] \vee data\_o[7] \vee data\_o[6] \vee data\_o[5] \vee data\_o[4]$
   $LNE[1] \leftarrow$
   $data\_o[14] \vee data\_o[11] \vee data\_o[10] \vee data\_o[7] \vee data\_o[6] \vee data\_o[3] \vee data\_o[2]$
   $LNE[0] \leftarrow$
   $data\_o[13] \vee data\_o[11] \vee data\_o[9] \vee data\_o[7] \vee data\_o[5] \vee data\_o[3] \vee data\_o[1]$
**end**

**return** $LNE[3:0]$

---

$$tw0 = LNE \qquad \text{(No shift, weight } = 2^0) \qquad (26)$$

$$tw1 = LNE \ll 1 \qquad \text{(1-bit left shift, weight } = 2^1) \qquad (27)$$

$$tw2 = LNE \ll 2 \qquad \text{(2-bit left shift, weight } = 2^2) \qquad (28)$$

$$tw3 = LNE \ll 3 \qquad \text{(3-bit left shift, weight } = 2^3) \qquad (29)$$

Each $tw_i$ represents $LNE$ multiplied by $2^i$, corresponding to the weight of bit $i$ in the binary representation.

**Conditional Summation Phase** The final product is computed by conditionally summing the shifted values based on the bits of $\alpha$:

$$tprod = \sum_{i=0}^{3} \left( \alpha[i] \ ? \ tw_i \ : \ 8'b0000\_0000 \right) \qquad (30)$$

$$prod = \$signed\$\{1'b0, tprod[7:4]\} \qquad (31)$$

**Mathematical Interpretation** This implementation effectively computes:

$$\alpha \cdot LNE \approx \frac{1}{16} \times (\alpha[3] \cdot 8LNE + \alpha[2] \cdot 4LNE + \alpha[1] \cdot 2LNE + \alpha[0] \cdot LNE)$$

where the final right-shift by 4 bits (selecting $tprod[7:4]$) and scaling by $\frac{1}{16}$ achieve the fractional multiplication.

B.2 RESOURCE ANALYSIS

To align with realistic hardware constraints, we further refine our encoder design for resource analysis. In particular, our system must co-exist with the analog front-end (AFE), whose silicon footprint is approximately $300\,\mu m \times 200\,\mu m$. To keep the digital compressor within a comparable or smaller area budget, we reduce the input window length to $T = 32$ samples at a sampling rate of $16\,kSps$, corresponding the algorithm process clock is 16kHz. This configuration corresponds to a 2 ms segment, which is sufficient to cover typical spike events.

The region of interest (ROI) is defined as the $\pm 1\,ms$ interval centered on the spike peak (i.e., 16 samples before and after). This choice both preserves the essential spike waveform information and allows implementation with a compact 32-sample shift register, requiring only lightweight control logic for peak-centered alignment.

Unless otherwise specified, all subsequent hardware-oriented measurements—including parameter count, bit width, buffer size, and operation count—are reported under this configuration. By grounding the encoder design in the AFE area constraint, the presented results reflect a feasible iBMI deployment scenario where extreme limits on memory, compute, and silicon area must be jointly satisfied.

**FPGA Implementation.** We first synthesized the proposed NSC module on a Xilinx xc7z020clg400-2 FPGA using Vivado. The results show that the NSC requires 161 Slice LUTs, 91 Slice Registers, and 54 occupied slices, with all LUTs used as logic. The measured dynamic power consumption is below 1 mW (reported as $< 0.001\,W$ by the tool), confirming the ultra-low power nature of the design.

**ASIC Synthesis.** To further evaluate silicon feasibility, we synthesized the entire digital processing chain (data storage, spike detection, and the NSC module) using Synopsys Design Compiler with the `scc018ug_uhd_rvt_ss_v1p62_125c_basic` standard-cell library. The total area is $34,678\,\mu m^2$, of which the NSC encoder itself occupies $11,465\,\mu m^2$. Timing analysis shows a slack of $62,488\,\mu s$ (MET), indicating that the design easily meets the target clock frequency(16kHz). The

power report indicates extremely low consumption: $2.60 \times 10^{-4}$ mW internal, $9.66 \times 10^{-6}$ mW switching, and $0.756\,\mu$W leakage, summing to a total of $1.03 \times 10^{-3}$ mW.

These results demonstrate that the NSC encoder not only meets strict FPGA resource limits but also achieves negligible area and power cost when mapped to a $0.18\,\mu m$ CMOS technology.

In future work, we will evaluate fully streaming in vivo scenarios.

## C    REPRODUCIBILITY STATEMENT

### C.1    CODE STATEMENT

The Python code will be released upon acceptance. In the double-blind review process, the code is available in the supplementary material. All written in Jupyter Notebook with a README file.

### C.2    DATASETS PROCESS

For all datasets, we adopted a unified preprocessing pipeline to obtain spike-centered signal segments. First, for those datasets that have raw signals without preprocessing, we applied a 300 Hz high-pass filter to remove low-frequency components. This step is standard in extracellular recordings and can be equivalently implemented in the analog front-end (AFE) hardware, thus not introducing additional digital computation overhead.

Second, candidate spike events were detected using an amplitude threshold. Following Donoho's principle Quian Quiroga (2009), the threshold was set within $[5, 50]$ times the median absolute deviation (MAD) of the filtered signal, using the same process as Chaure et al. (2018), ensuring robust spike detection while suppressing noise fluctuations. To avoid detecting overlapping spikes within the refractory period, we eliminated consecutive events that were closer than 3 ms, corresponding to typical neuronal firing constraints.

Third, around each detected spike, we extracted a fixed-length window of 128 samples, where the spike location was centered. The extracted segments were then quantized into signed 8-bit integers within the range $[-128, 127]$. This quantization step ensures hardware compatibility, as the input to our encoder consists entirely of integer-valued samples suitable for efficient on-chip implementation.

Through this process, we obtained consistent spike-aligned segments across multiple datasets, all represented in the same integer domain. These preprocessed datasets serve as the input for training and evaluating our proposed neural signal codec.

Now we will describe the data processing procedures for each dataset in sequence.

**QU Dataset**    The QU dataset provides raw data in .mat files, which include ground truth spike indices and their corresponding classes. Therefore, we only quantized and scaled the data to 8-bit resolution, after which we extracted the signal segments and their nonlinear energy operator (NEO) components based on the provided indices.

**GC Dataset**    This dataset provides raw recordings across 256 channels along with corresponding trigger files (containing spike indices) in '.npy' format. Data loading instructions are available on the official website. In our processing, we used only the single channel specified in the trigger files to construct the dataset. No filtering was applied. Similar to the procedure for the QU dataset, the data from the selected channel were quantized and scaled to 8 bits, after which signal segments and NEO components were extracted using the provided indices.

**hc1**    For this dataset, we first applied a 300 Hz highpass filter. The recordings comprise six channels, with the $6-th$ channel containing the juxtacellular recording, which served as the ground truth for spike indexing. Spikes detected on the juxtacellular reference channel were concurrently applied to the data channels (Channels 2, 3, 4, and 5). Since these extracellular channels recorded activity from the same cell, the spike classes across them are identical; the primary distinction lies in their respective response amplitudes. Following this procedure, the data was similarly quantized to 8 bits.

**NP** This dataset includes recordings from 384 channels. The official website provides spike-sorting results, so we directly used these outputs. The data were only quantized to 8-bit integer format for subsequent analysis.

**MIT-BIH** Similarly, for the MIT-BIH dataset, no filtering was applied. The datasets provide raw signals along with annotated indices. We directly extracted the corresponding signal segments using these indices and then quantized the data to 8-bit resolution.

In the ablation study, we utilized the QU Difficult1Noise02 dataset and the NP channel 1 data. For the comparative experiments, we constructed mixed datasets to evaluate model performance under more realistic and varied conditions:

For the GC dataset, recordings from sessions 20170622, 20170623, and 20170629 were combined to form a mixed dataset containing three spike classes. For the HC1 dataset, recordings d533101 and d561106 were merged to create a two-class mixed dataset.

Noticed that since both GC and HC1 are extracellular recordings from individual neurons, each original recording contains only one type of spike. The mixed datasets were therefore constructed by combining multiple channel recordings to simulate multi-spike-class scenarios.

### C.3 Training Settings

All experiments were conducted on a workstation equipped with one NVIDIA GeForce RTX 5090 (32 GB) GPU, an Intel Xeon Platinum 8470Q CPU (25 vCPUs), and 90 GB of system memory. The code was implemented in Python 3.12 (on Ubuntu 22.04) using PyTorch 2.8.0 and CUDA 12.8.

For reproducibility, all models were trained with random seeds fixed from 1 to 5, managed via the $seed\_everything$ function from PyTorch Lightning. We used the AdamW optimizer with its default parameters and a global learning rate of $1e-3$ for all models, without employing a learning rate scheduler. A weight decay of $1e-4$ was applied to all non-quantized parameters, and gradient clipping with a maximum norm of 10 was used during training.

The model architecture was consistent across experiments: input and reconstruction dimensions were set to 128, with a latent space size of 4. Each model was trained for 100 epochs using mean squared error (MSE) as the default loss function, unless otherwise specified in the main text. A validation set was used for monitoring performance, but no early stopping was applied. Parameter initialization followed the default methods of the framework, without specific modifications.

Regarding data preparation, no additional normalization or data augmentation was applied. The input signals were directly used after 8-bit integer quantization.

**Specified Initialization** For our proposed Neural Spike Coder (NSC), the scaling factor $\alpha$ was initialized uniformly in $[0.25, 0.35]$ using $\alpha \sim \mathcal{U}(0,1) \cdot 0.1 + 0.25$. The shift parameter $\beta$ was initialized to zero, and the quantization threshold $\gamma$ was set to 0.5 initially. The window boundaries were determined by averaging the input range into equal intervals.

For the baseline models, including AE_QINT8, AE_Q1P4, and VQ VAE, all quantized parameters were uniformly initialized between the minimum and maximum values of the corresponding input data range.

### C.4 Result Process

For the clustering evaluation, we employed the K-means algorithm with the random seed fixed to 42 to ensure reproducibility. To accurately assess the clustering performance against the ground truth labels, we resolved the label assignment ambiguity using the Hungarian matching algorithm. This method finds the optimal one-to-one mapping between predicted clusters and true classes by maximizing the overall alignment between the two sets of labels. The remapped labels were subsequently used to compute all clustering metrics reported in the study.

## D  ETHICS STATEMENT

All datasets used in this study are publicly available for research purposes under open-access licenses. Our experiments involved only secondary analysis of existing data and did not involve any new data collection from human subjects. Therefore, no ethical approval was required for this work.

To access the original datasets :

QU (CC BY 4.0): `https://figshare.le.ac.uk/articles/dataset/Simulated_dataset/11897595?file=21819066`

GC (CC BY 4.0): `https://zenodo.org/records/1205233#.XMH886xKjCI`

hc1 (CC BY 4.0): `https://crcns.org/data-sets/hc/hc-1`

NP (CC BY-NC 4.0): `https://rdr.ucl.ac.uk/articles/dataset/Recording_with_a_Neuropixels_probe/25232962/2?file=44571832`

MIT-BIH (ODC-By): `https://physionet.org/content/mitdb/1.0.0/`

## E  THE USE OF LARGE LANGUAGE MODELS (LLMS)

In the preparation of this manuscript, the authors utilized the large language models (ChatGPT and Deepseek) as an auxiliary tool to enhance the writing and editing process. The model was employed specifically for text polishing, grammar correction, and improving the fluency and clarity of certain passages in the manuscript.

It is important to note that all scientific content, including the core ideas, theoretical framework, experimental design, results, and conclusions, originated solely from the authors. The LLM did not contribute to the intellectual substance of the work, nor was it used to generate any scientific insights, data, or interpretations.

The authors have reviewed and edited all AI-assisted content and take full responsibility for the entire work, including its accuracy and integrity.

The use of the LLM was guided by and under the continuous supervision of the authors, adhering to the principles of transparency and responsible AI use in academic research.

## F  SUPPLEMENT DATA

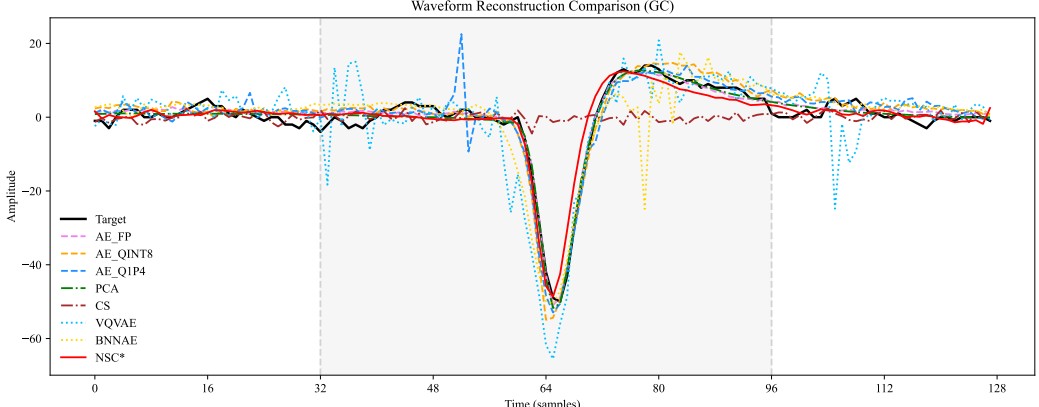

Figure 8: Reconstructed (GT) Testset Index 0 Waveform (seed 1)

**Reconstructed Wave Visualization**

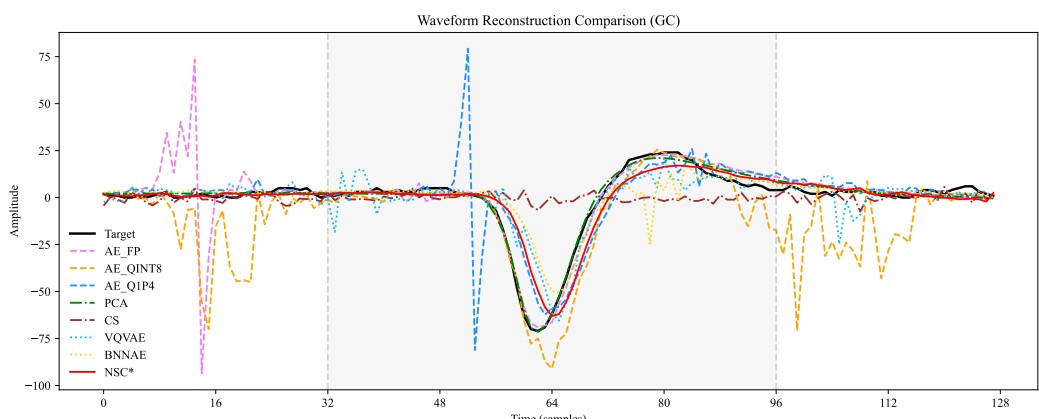

Figure 9: Reconstructed (GT) Testset Index 1 Waveform (seed 1)

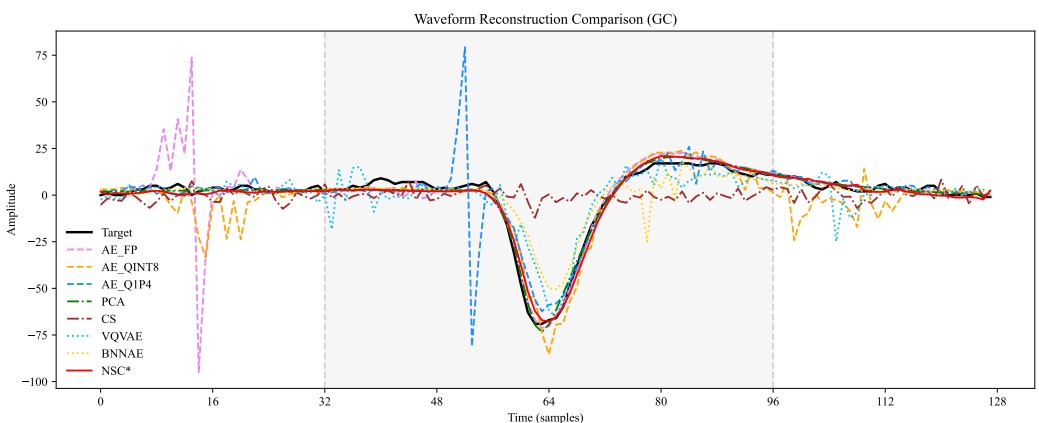

Figure 10: Reconstructed (GT) Testset Index 2 Waveform (seed 1)

To provide qualitative evidence of reconstruction fidelity, we visualize several example waveforms from the test sets. Figures 8–10 show the first three test samples from the GT dataset (seed 1), while Figures 11–13 show the corresponding first three samples from the HC1 dataset (seed 1). Each figure compares the reconstructed waveform with the ground-truth spike. These examples illustrate that our proposed NSC encoder preserves key spike morphology and achieves high reconstruction quality across different datasets.

# G   Q&A

## APPENDIX B: SPIKE ALIGNMENT Q&A

### B.1 SPIKE TIMING ALIGNMENT IMPLEMENTATION

**Q.1: Why are all spikes aligned to the center of the time window during testing? Does this approach introduce additional computational overhead?**

**A.1: This alignment does not introduce any computational overhead**. It can be achieved through a simple shift register architecture.

Let's take a 128 waveform as an example. The key insight is that we use a 64-element shift register ([0: 63]) as a delay line. Each clock cycle, we:

1. Shift in one new data sample at the input (position 63, tail)

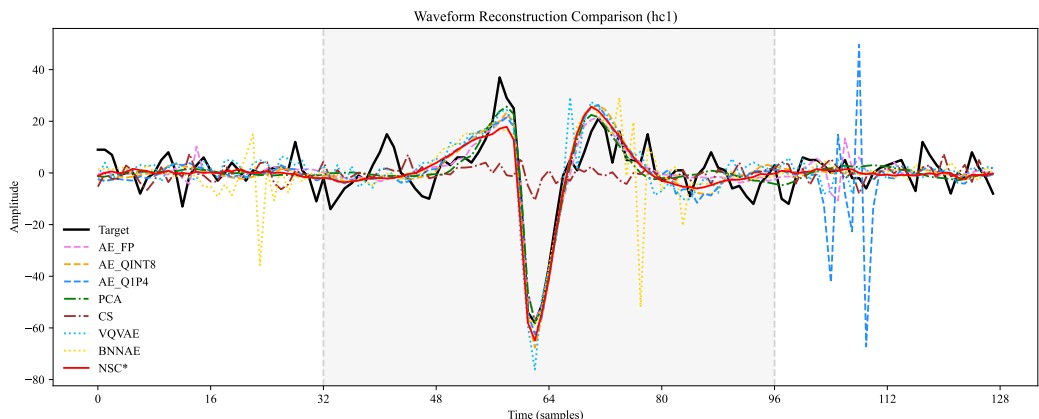

Figure 11: Reconstructed (hc1) Testset Index 0 Waveform (seed 1)

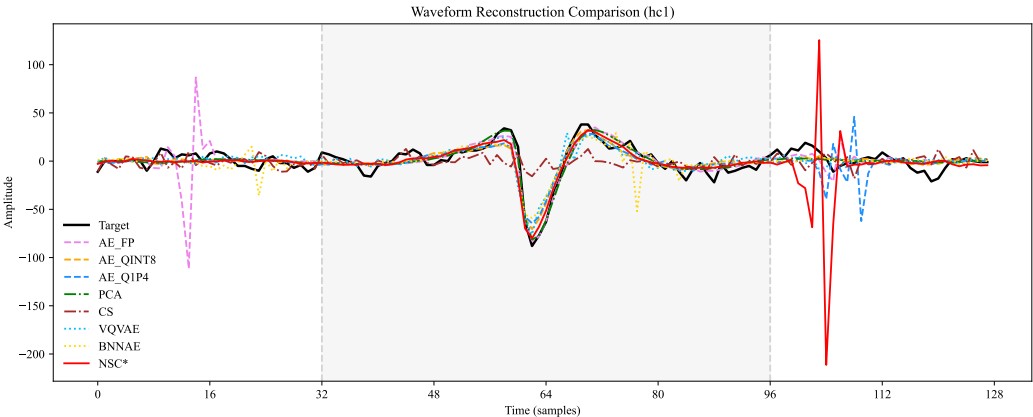

Figure 12: Reconstructed (hc1) Testset Index 1 Waveform (seed 1)

2. Shift out the oldest sample at the output (position 0, head)

3. Perform spike detection on the newest sample at position 63

When a spike is detected at position 63, the current shift register contains the first 64 samples of the 128-sample window. The sample at position 0 was actually read 64 clock cycles ago, representing the historical data. We then simply continue collecting data for another 64 clock cycles to complete the 128-sample window.

**Timing Analysis:**

- **At detection time (t=0):** Position 0 contains data from 64 cycles ago

- **After 64 more cycles (t=64):** Position 0 will contain the spike detected sample; now we have read half the waveform.

- **Result:** The spike is naturally centered in the 128-sample window

This approach requires only basic shift register operations—no complex addressing, no additional buffers, and no computational correction. The center alignment emerges naturally from the timing relationship between data entry, detection point, and continued data collection.

**Q.2: Why is the design primarily focused on the ROI region, and is this scientifically justified?**

**A.2: Yes, the focus on a specific Region of Interest is strongly grounded in established biological research practices**. In neural signal analysis, the critical features of an action potential, or spike, are typically contained within a standardized time window around its

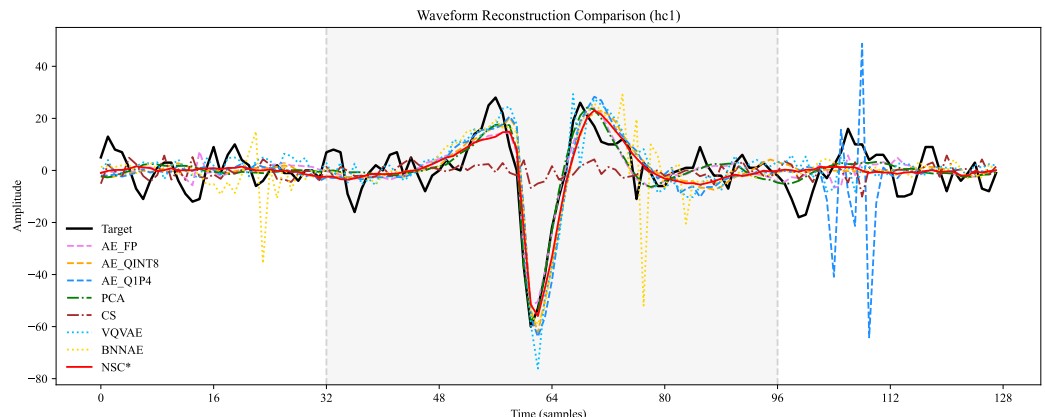

Figure 13: Reconstructed (hc1) Testset Index 2 Waveform (seed 1)

peak. Conventionally, researchers analyze a segment spanning approximately 0.5 milliseconds before the peak to 1.0 milliseconds after it. This window captures the essential rising and falling phases of the spike waveform, which are crucial for neuron identification, sorting, and analysis Toosi et al. (2021). Our design captures a 64-sample window centered on the detected spike. At a sampling rate of 32 kilohertz, which is considered high resolution for electrophysiological data, this window duration is exactly 2.0 milliseconds. This provides 1.0 milliseconds of data on either side of the central detection point. Therefore, our chosen ROI not only meets but exceeds the conventional research requirement of 1.5 milliseconds, ensuring that the complete, scientifically relevant waveform morphology is captured without unnecessary data overhead.

**Q.3: Why is it necessary to minimize parameters? Couldn't conventional lightweight neural networks be used?**

**A.3:** The imperative for an extremely low parameter count stems directly from hardware constraints, not just for algorithmic simplicity. The choice is dictated by the need for area efficiency and to maintain a balanced design between the analog and digital domains.

In the 180nm semiconductor technology node, the physical area of a single 1-bit register is approximately 45 $\mu m^2$. In contrast, a typical analog front-end circuit, which includes components like amplifiers and an Analog-to-Digital Converter, occupies an area of roughly 300 $\mu m$ by 200 $\mu m$.

Consequently, the digital logic's area must be designed to be commensurate with that of the analog section to achieve a balanced overall system. A parameter-dense model, even those labeled as "lightweight" in software terms, would necessitate thousands of registers and computational elements. Crucially, the area figures mentioned pertain to a single pixel or channel. When integrated into a large-scale array containing hundreds or thousands of such units, the total silicon area would become immense. If the digital block for each unit were significantly larger than its analog counterpart, the aggregate area disparity would be magnified across the array, resulting in a severely unbalanced chip. Therefore, conventional lightweight networks are unsuitable.

