# OpenReview forum: "A Neural Signal Codec with Resource Efficient Encoder for Implantable Brain Machine Interface Systems"
_ICLR.cc/2026/Conference — Submitted to ICLR 2026_

### Official Review · Reviewer_AAjh · 2025-10-31

**Soundness:** 2
**Presentation:** 2
**Contribution:** 2
**Rating:** 2
**Confidence:** 5

**Summary:**

This paper presents a Neural Signal Codec (NSC) designed specifically for implantable brain-machine interface (iBMI) systems. The proposed method features a highly resource-efficient encoder with only 124-bit parameters that operates without multiplications, making it suitable for on-chip deployment. The encoder employs a dynamic weight generation mechanism for parameter sharing within windows and uses 4-bit quantization. On the decoder side, a conventional CNN architecture is combined with a novel Energy-Aware Loss (EAL) that emphasizes signal energy-intensive regions. The authors claim a 97% parameter reduction compared to conventional FC-based autoencoders with INT8 quantization. Extensive evaluations across multiple neural signal datasets demonstrate the method's ability to preserve biological features while maintaining high compression ratios (32:1). Additional experiments on ECG signals suggest potential generalization to other biosignal domains.

**Strengths:**

Novel and Hardware-Oriented Design:​​ The paper addresses a critical challenge in iBMI systems - extreme resource constraints - with a carefully designed multiplication-free encoder. The dynamic weight generation mechanism and parameter sharing scheme represent innovative approaches to minimize storage requirements while maintaining functionality.
​​Comprehensive Evaluation:​​ The authors evaluate their method on multiple datasets (QU, GC, hc1, NP) with both synthetic and real neural signals, demonstrating robustness across different recording conditions. The inclusion of downstream clustering analysis provides valuable insights into practical utility.
​​Strong Resource Efficiency:​​ The 124-bit parameter footprint represents a significant advancement for implantable applications.

**Weaknesses:**

Inadequate Related Work Coverage:​​ The paper lacks comparison with learned compression methods from other domains, particularly learned image compression techniques (e.g., VQ-VAE, transform coding models) that share similar encoder-decoder architectures and discrete representation learning concepts. This omission makes it difficult to assess the novelty of the overall framework.
​​Unclear Experimental Conditions:​​ The description of baseline implementations is insufficient. Equation 7 indicates the NSC produces discrete codes, but it's unclear whether compared methods (especially AE_FP32) also produce discrete representations. If not, the comparison becomes unfair. Additionally, the paper doesn't specify if all methods were compared at equivalent bitrates, which is essential for meaningful compression performance evaluation.
​​Incomplete Results Interpretation:​​ Table 4 shows significant performance gaps in clustering metrics (F1 score: 0.98 for AE_FP32 vs. 0.72 for NSC), but the paper doesn't adequately discuss whether this level of performance degradation is acceptable for practical neural signal analysis. The biological implications of this reduction need more thorough discussion.

**Questions:**

Please address the questions in the weaknesses part.

---

> ### Author Response · Authors · 2025-11-17
> **Response to Reviewer AAjh**
>
> We thank the reviewer for the time spent on reviewing this paper and the valuable comments. We have carefully read through the comments and respond to each comment as below.
>
> **Weakness:**
>
> **Comment 1:**   Inadequate Related Work Coverage: The paper lacks comparison with learned compression methods from other domains, particularly learned image compression techniques (e.g., VQ-VAE, transform coding models) that share similar encoder-decoder architectures and discrete representation learning concepts. This omission makes it difficult to assess the novelty of the overall framework.
>
> **Response 1:**  We thank the reviewer for pointing out related works. Our experiments already evaluate a wide spectrum of compression techniques, covering conventional, learned, and quantized approaches, including AE-FP32, AE-INT8, AE-INT1.4, PCA, compressive sensing, VQ-VAE (line 274), and binary neural network (Table 3). We demonstrated potential models for both continuous and discrete, as well as classical linear and NN-based methods.
>
> And it is important to note that many cross-domain learned compression frameworks can’t be meaningfully compared in the implant scenario, such as large VQ-VAE, image, audio codecs.  In our fabrication process (180 nm), the digital area is approximately 60000 \mu m^2, with single 1-bit register occupies ~45 \mu m^2.  Giving total ~1300 bits ideally for all digital storage, including parameters, buffers, intermediate registers, params, and the routing wire. Vanilla single-layer MLP even exceeds the demands.
>
> As Table 3 shows, our method (NSC) only uses 124 bits in parameters. However, typical learned codecs need orders of magnitude more—codebook storage alone would exceed the total implant memory by 10³–10⁵ times. Evaluating these large models under implant constraints wouldn’t be informative, their parameter size alone blows the silicon budget, even before considering compute logic. We’ll clarify these areas and memory constraints in the revision.
>
> **Comment 2:**   Unclear Experimental Conditions: The description of baseline implementations is insufficient. Equation 7 indicates the NSC produces discrete codes, but it's unclear whether compared methods (especially AE_FP32) also produce discrete representations. If not, the comparison becomes unfair. Additionally, the paper doesn't specify if all methods were compared at equivalent bitrates, which is essential for meaningful compression performance evaluation.
>
> **Response 2:**  We have to clarify the experimental conditions and bit rate normalization. As stated in the manuscript (lines 1326–1330), all experiments used the same input and reconstruction dimensions (T = 128) and the same latent dimensionality (d = 4), which fixed a 32× compression factor (128 samples × 8 bits = 1024 input bits to 4 latent values = 32 bits).
>
> “AE_FP32” denotes full-precision model parameters (32-bit float points) and serves only as an upper-bound reference for reconstruction quality, does not imply a different transmitted bitrate. And all the models’ name refers to the parameter description (lines 268-276).
>
> For the unfair claim, we make no artificial constraint for the latent outputs by default. All baselines compute latent values in floating point. While our model happened to be integers, this is given by the structural consequence of the encoder hardware architecture. In other words, NSC naturally produces integer latent codes in computation, whereas other baselines produce floating-point outputs.
>
> For the settings:
>
> Inputs: 128 lengths (dims) sample points, 8-bit signed integer (line 289).
>
> Model params: named in the models, shown in Table 3 (lines 432-444, last column).
>
> Latent value: 4 lengths (dims), no artificial constraint, floating points, our model produces the integer due to the designed hardware computation structure.

---

> > ### Comment · Reviewer_AAjh · 2025-11-27
> > **Response to authors' rebuttal**
> >
> > Thank the authors for providing the rebuttal. It partly addresses my concerns but fails to resolve all. Let me make my points clearer:
> > - "Comment 1: Inadequate Related Work Coverage": I now better understand the distinction of the present study in contrast to the prior arts. It would be much better if the authors could discuss the novelty with respect to the wider context of the literature of learned data compression.
> > - "Comment 2: Unclear Experimental Conditions": I still believe the comparison is not fair. "NSC naturally produces integer latent codes in computation, whereas other baselines produce floating-point outputs." This makes the comparison unfair because the latent codes are either discrete (NSC, also VQ-VAE if I understand correctly) or continuous (other baselines). While the authors overlook this difference by claiming that "Latent value: 4 lengths (dims), no artificial constraint, floating points, our model produces the integer due to the designed hardware computation structure," this cannot persuade me. The difference between discrete and continuous latent representations is the key difference between VQ-VAE (and its variants) and normal VAEs. Nonetheless, I believe the authors have clarified the experimental settings, which disclose the unfair settings. I would urge the authors to discuss that a fairer comparison is difficult due to the lack of baselines.

---

> ### Author Response · Authors · 2025-11-17
> **Response to Reviewer AAjh**
>
> **Comment 3:**    Incomplete Results Interpretation: Table 4 shows significant performance gaps in clustering metrics (F1 score: 0.98 for AE_FP32 vs. 0.72 for NSC), but the paper doesn't adequately discuss whether this level of performance degradation is acceptable for practical neural signal analysis. The biological implications of this reduction need more thorough discussion.
>
> **Response 3:**  Our goal is not to maximize the global clustering scores and we expect to find an implantable encoder that balance hardware limits and downstream utility.
>
> NSC is optimized for region-of-interest (ROI) fidelity via the Energy-Aware Loss (EAL) by focusing on high-energy temporal segments (spikes). Thus, reconstruction and clustering of ROIs remain extremely strong whereas a reduction in some full-window metrics. This behavior is evident in Table 4: NSC (EAL) achieves near-perfect ROI metrics on GC (ROI F1 = 1.00±0.00, ARI = 0.99±0.01, NMI = 0.98±0.01) and correspondingly high scores on hc1. These findings suggest that NSC preserves spike morphology—the biologically meaningful part of the signal—with high reliability.
>
> The lower global F1 is an intentional bitrate, area, and fidelity trade-off, and not due to a failure of reconstruction. Global metrics consider low-energy background samples on par with spikes and noise; further improvements of these metrics would have required extra model capacity incompatible with implantable area and power budgets. AE_FP32 is also a floating-point bound, whose parameter size and compute costs exceed the implantable feasibility by orders of magnitude (see Table 3). NSC achieves its performance while remaining within the strict on-chip budgets.
>
> We appreciate the reviewer’s suggestions — they will improve clarity and practical relevance. We will add the mentioned results in the revision.

---

> > ### Comment · Reviewer_AAjh · 2025-11-27
> > **Response to authors' rebuttal**
> >
> > - "Incomplete Results Interpretation": I feel better now because the authors clarify that NSC is optimized for ROI fidelity rather than treating the entire signal equally. However, this leads to a new question: What will the baselines perform if they are also optimized for ROI fidelity? I think this is a critical question to isolate the real benefit of the method from the optimization target.
> > In summary, I am still not fully convinced by the authors' rebuttal but I would say that the provided details strengthen the paper. I would like to raise my score to 4.

---

> > > ### Author Response · Authors · 2025-11-28
> > > **Second Response to Reviewer AAjh**
> > >
> > > Thanks for your response and comments, and we will respond to each comment as follows.
> > >
> > > **Comment :**   Unclear Experimental Conditions & Incomplete Results Interpretation
> > >
> > > **Response :** We thank the reviewer and provide two clarifications and additional experiments to ensure fair comparison.
> > >
> > > (1 ) Adding a full-precision NSC baseline for fairness. To fully isolate the effect of quantization and ensure fair comparison, we additionally trained NSC-FP32, a full-precision variant of NSC where all encoder parameters (\alpha, \beta and \gamma) operate in floating point while clamped to the same range as discrete presentation, no quantization or integer constraints are applied, and the decoder is the same as the original NSC. Details will be reported in the revision.
> > >
> > > This provides a directly comparable baseline to AE_FP32 etc., and quantifies how much of NSC’s behavior is due to the architecture rather than quantization. Including NSC-FP32 ensures that the comparison is completely fair — and that NSC is not advantaged nor disadvantaged by integer-only operations.
> > >
> > > (2) Unified training under MSE and EAL-G
> > > To address potential concerns about training mismatch, we additionally re-trained all models using both MSE and EAL-G losses, and report results on the GC datasets below (hc1 datasets will be included in the revision).
> > > Since CS and PCA do not rely on learnable reconstruction losses, we omit them in this table to avoid misleading comparisons.
> > >
> > >
> > >
> > > | **Setting**   | **Train with MSE**  |             |                  |
> > > |---------------|---------------------------------|---------------------------------|-----------------------------------|
> > > |               | PSNR$_{\text{FULL/ROI}}$ $\uparrow$   | SNDR$_{\text{FULL/ROI}}$ $\uparrow$       | NRMSE$_{\text{FULL/ROI}}$ $\downarrow$     |
> > > | AE_FP32          |23.66$\pm$1.21/25.97$\pm$2.85 | 8.38$\pm$1.21/13.50$\pm$2.83 | 0.08$\pm$0.01/0.06$\pm$0.02|
> > > | AE_INT8          |22.23$\pm$1.32/22.29$\pm$1.93 | 6.94$\pm$1.32/9.82$\pm$1.92 | 0.09$\pm$0.01/0.09$\pm$0.02|
> > > | AE_INT1.4       |18.96$\pm$2.00/17.95$\pm$3.35 | 3.67$\pm$1.98/5.48$\pm$3.34 | 0.13$\pm$0.02/0.16$\pm$0.05|
> > > | VQ VAE            |18.63$\pm$1.59/21.76$\pm$4.06 | 3.34$\pm$1.60/9.29$\pm$4.07 | 0.13$\pm$0.02/0.10$\pm$0.05|
> > > | BNN AE 	 |17.60$\pm$1.22/17.19$\pm$1.43 | 2.31$\pm$1.23/4.72$\pm$1.41 | 0.14$\pm$0.02/0.15$\pm$0.02|
> > > | NSC_FP32	 |22.70 $\pm$ 0.96 / 23.40 $\pm$ 1.55 | 7.41 $\pm$ 0.94 / 10.93 $\pm$ 1.56 | 0.08 $\pm$ 0.01 / 0.08 $\pm$ 0.02|
> > > | NSC		 |22.87$\pm$2.43/23.01$\pm$1.93 | 7.58$\pm$2.42/10.54$\pm$1.94 | 0.09$\pm$0.02/0.10$\pm$0.03|
> > >
> > >
> > > | **Setting**   | **Train with EAL-G**|            |                      |
> > > |---------------|---------------------------------|---------------------------------|-----------------------------------|
> > > |               | PSNR$_{\text{FULL/ROI}}$ $\uparrow$        | SNDR$_{\text{FULL/ROI}}$ $\uparrow$       | NRMSE$_{\text{FULL/ROI}}$ $\downarrow$     |
> > > | AE_FP32         | 26.30 $\pm$ 1.17 / 29.94 $\pm$ 0.75 | 11.01 $\pm$ 1.18 / 17.48 $\pm$ 0.76 | 0.08 $\pm$ 0.03 / 0.04 $\pm$ 0.00 |
> > > | AE_INT8          |22.67 $\pm$ 1.85 / 24.77 $\pm$ 1.03 | 7.38 $\pm$ 1.85 / 12.30 $\pm$ 1.03 | 0.20 $\pm$ 0.21 / 0.07 $\pm$ 0.01|
> > > | AE_INT1.4       |18.86 $\pm$ 1.97 / 21.17 $\pm$ 1.48 | 3.57 $\pm$ 1.96 / 8.70 $\pm$ 1.46 | 0.24 $\pm$ 0.19 / 0.11 $\pm$ 0.01|
> > > | VQ VAE            |18.63 $\pm$ 1.59 / 21.76 $\pm$ 4.06 | 3.34 $\pm$ 1.60 / 9.29 $\pm$ 4.07 | 0.13 $\pm$ 0.02 / 0.10 $\pm$ 0.05|
> > > | BNN AE 	 |8.99 $\pm$ 5.25 / 19.97 $\pm$ 1.04 | -6.30 $\pm$ 5.24 / 7.50 $\pm$ 1.04 | 0.49 $\pm$ 0.22 / 0.12 $\pm$ 0.01|
> > > | NSC_FP32	 |24.18 $\pm$ 1.41 / 26.05 $\pm$ 0.08 | 8.89 $\pm$ 1.39 / 13.59 $\pm$ 0.08 | 0.10 $\pm$ 0.04 / 0.05 $\pm$ 0.00|
> > > | NSC		 |21.12$\pm$6.38 / 25.70$\pm$0.21 | 5.83$\pm$6.36 / 13.23$\pm$0.21 | 0.19$\pm$0.16 / 0.06$\pm$0.00|
> > >
> > > *Notes* : NSC_FP  Encoder Params are 880bits
> > >
> > >
> > > Summary: (1) As expected, training with EAL-G improves the ROI fidelity of baselines across almost all models. (2) NSC-FP32 shows that it is the architecture, not quantization, that accounts for the reconstruction performance.

---

### Official Review · Reviewer_TXDs · 2025-11-01

**Soundness:** 2
**Presentation:** 2
**Contribution:** 2
**Rating:** 4
**Confidence:** 3

**Summary:**

The paper proposes a neural signal codec through an encoder-decoder architecture, where the encoding is done on-chip or on another possible edge device. The authors propose a simple mechanism to extract regions of interest that can be decoded through a multilayer CNN trained with a proposed energy aware loss function. The authors compare their method on various datasets, showing their performance on the full signal and specific regions of interest.

Claimed Contributions:

- Proposed an encoder-decoder architecture for signal reconstruction in a constrained resource setting
- Introduced a new loss for spike reconstruction

**Strengths:**

- developed a solution that focuses on efficiency and outperforms other methods in a low-constrained setting.
- The motivation for their approach is well grounded in theory and experiments

**Weaknesses:**

1. Lack of comparison to the relevant literature. Within the relevant work, the authors show some of the more recent work, yet they do not evaluate or do not tie the work to their method. The work introduced in the “On-Chip Neural Signal Compression” section is briefly mentioned, but the direct relevance is not shown.
2. Poor presentation: The graphics and tables in the paper are barely visible and of low quality. The images have poor resolution, with Figure 3 being impossible to view. Additionally, Tables 4 and 5 are difficult to read, as there is no visual indication of what the reader should focus on, given the plethora of data presented.
3. The design choices and the algorithm are not motivated well:
- a) In the “Ablation Studies”, the authors point out that “The general trend across datasets is that performance improves with larger w”, but later claim that “larger window counts increase storage without consistent gains across datasets”, showing inconsistencies in design choices and reasoning.
- b) Moreover, the increase in memory is not quantified.
- c) Moreover, some parts of the algorithm are not explained. In the energy-aware loss, the authors mention smoothing e (line 248) and then smoothing it again with Equation 8.
4. No ablation nor discussions on the decoder: While I acknowledge the focus of the paper on the encoder architecture, the decoder architecture and the design choices are not documented.
5. Minor writing mistakes: Some repetition and mistakes that break the flow of the paper. Line 51 with the repetition and Line 313 with the redefinition of the NP channel.

**Questions:**

I have one central question: What considerations have been made when it comes to the design of the decoder architecture?

---

> ### Author Response · Authors · 2025-11-17
> **Response to Reviewer TXDs**
>
> We thank the reviewer for the time spent on reviewing this paper and the valuable comments. We have carefully read through the comments and respond to each comment as below.
>
> **Weakness:**
>
> **Comments 1:** Lack of comparison to the relevant literature. Within the relevant work, the authors show some of the more recent work, yet they do not evaluate or do not tie the work to their method. The work introduced in the “On-Chip Neural Signal Compression” section is briefly mentioned, but the direct relevance is not shown.
>
> **Response 1:** We agree that the connection of the NSC within the general compression literature enhances clarity. As concerns the related work explained within “On-Chip Neural Signal Compression”, we are grateful for the chance to clarify the connection to the proposed work.
>
> High-loss algorithms such as basic thresholding, spike detection and sorting. These algorithms involve the transmission of very sparse information such as the occurrence of the spikes themselves or the timestamps of the spikes. These algorithms involve the discarding of the morphological information of the waveforms. As the contribution of the current work focuses on low-loss waveform reconstruction algorithms that retain information related to the morphological characteristics of the spikes (an important requirement for various iBMI decoding frameworks), these algorithms are not considered to be directly similar.
>
> Low-loss algorithms such as Autoencoders and PCA. These techniques are designed to retain the waveform shapes. While they are computationally and parametrically intensive, using multipliers, dense layers, and floating-point operations. These are not very useful for implant devices, where the computational resources are very limited. The scales of the parameters are already shown in Table 3.
>
> What we already evaluated has shown in Table 3. Including baselines: AE-FP32, AE-INT8, AE-INT1.4, PCA, CS, VQ-VAE, and a BNN-AE. These baselines cover continuous & discrete param models, classical mathematical methods, and extreme quantization.
>
> Many cross-domain learning codecs do not constitute meaningful implementation baselines. Modern large VQ-VAE variants, transformer-based codecs, and others all depend on components that are infeasible on tight implant hardware, which requires large codebooks, entropy encoders, and dense matrix multiplies. NSC requires ∼124 bits of parameter storage; conventional learned codecs exceed the on-chip budget by multiple orders of magnitude. Thus, their evaluation as implant baselines would be neither fair nor informative for the target application.
>
> **Comments 2:** Poor presentation: The graphics and tables in the paper are barely visible and of low quality. The images have poor resolution, with Figure 3 being impossible to view. Additionally, Tables 4 and 5 are difficult to read, as there is no visual indication of what the reader should focus on, given the plethora of data presented.
>
> **Response 2:** We apologize for the low-resolution figures in the submitted PDF. We will replace all figures with high-resolution images, enlarge fonts, and reformat Tables 4–5 to highlight the primary comparisons. Figure 3 (loss curves) will be replotted with larger markers and axis labels for clarity.
>
> **Comments 3(a) & (b):** a) In the “Ablation Studies”, the authors point out that “The general trend across datasets is that performance improves with larger w”, but later claim that “larger window counts increase storage without consistent gains across datasets”, showing inconsistencies in design choices and reasoning. b) Moreover, the increase in memory is not quantified.
>
> **Response 3(a) & (b):**	We thank the reviewer for pointing out the contradiction. The two statements in the paper are both correct but describe different aspects of the same trade-off. 1. Performance improves with larger w. This refers to absolute reconstruction quality: adding windows increases representational flexibility and thus tends to improve PSNR/ROI metrics. 2. Larger window counts increase storage without consistent gains across datasets, which refers to the marginal utility: the performance gain per additional window diminishes and is dataset-dependent, while encoder storage increases linearly.
>
> For the computation of the storage. Here we let the latent dim be, and we totally have windows. Thus, the encoder params can be computed with:
>
> 	Enc_bits = alpha + beta + gamma + window_boundaries
> 		= 4 * w * d + 4 * w * d + 4 * w + 8 * (w-1)
> 		= (8d + 12) w – 8
>
> For our design with d = 4 (32 times compression) and w=3, the total param bits are 124bits, which is consist to the Table 4. And based on the function, the params grow linear with the window count. Each window addition will increase ~2000 \mu m^2\ for param storage, now total area is ~30000 \mu m^2. In addition, intermediate variables and combinatorial logic are also required for cooperation.

---

> ### Author Response · Authors · 2025-11-17
> **Response to Reviewer TXDs**
>
> **Comments 3(c):**  The clarification for EAL smoothing
>
> **Response3(c):** It is correct for the review to point out that EAL involves two stages operating on the energy profile, but these two steps serve different purposes.
>
> The first smoothing, shown in line 248, applies a light Gaussian filter to the raw NEO curve. This removes most of the sampling noise and glitches introduced by the NEO operator, generating a stable energy envelope. Figure 3 shows the importance of this step using the code between lines 378 and 386.
>
> Equation (8), line 250, does not smooth the energy again but instead fits a probabilistic weighting function (Gaussian/Laplace/Cauchy shape) using the smoothed energy as input; this step will produce the final attention weights over the window, emphasizing high-energy regions in the reconstruction.
>
> We are sorry for the unclear expression. The pipeline is: raw NEO → denoised energy envelope → fitted attention distribution.
>
> **Comments 5:** Minor writing errors
>
> **Response 5:** Thanks for your careful reading and patience. We apologize for the writing errors and we will correct repetitions and line errors and proofread the manuscript to improve flow.
>
> **Questions:**
>
> **Comments & Weak 4:** Decoder Design
>
> **Response :** We apologize for not including the decoder architecture figure in the submitted PDF. The decoder is a fixed, lightweight CNN-with-residual-blocks. And we will add the missing decoder diagram and a detailed architectural table in the revision. The structure of the decoder is: (latent dim is z 4-dim). With settings of:
>
> | **Stage**   | **Layer**  |**Output Shape**|
> |---|---|-----|
> | FC| Linear(4->512), reshape|(B, 4, 128)|
> | Down1 | Conv1D, ResBlock |(B, 64, 64)|
> | Down2 | Conv1D, ResBlock |(B, 128, 32)|
> | Down3 | Conv1D, ResBlock |(B, 256, 16)|
> | Up1 | ConvTrans, ResBlock |(B, 128, 32)|
> | Up2 | ConvTrans, ResBlock |(B, 64, 64)|
> | Up3 | ConvTrans, ResBlock |(B, 32, 128)|
> | Out | Conv1D |(B, 1, 128)|
> Our work primarily focuses on the encoder, which is the main source of innovation. Accordingly, the decoder is designed with a minimal yet sufficient capacity — its goal is simply to reconstruct the waveform reliably, without dominating the overall learning process. A highly expressive decoder could easily overfit or mask the differences between encoders, making it unclear whether improvements come from the encoder or the decoder. To ensure a fair comparison, all methods in our study share the same base decoder architecture.
> And for the ablation of the designed decoder, it is as follows:
>
> | **Setting**   | **Syn (QU D1N2)**    |             |                  |
> |---------------|---------------------------------|---------------------------------|-----------------------------------|
> |               | PSNR$_{\text{FULL/ROI}}$ $\uparrow$   | SNDR$_{\text{FULL/ROI}}$ $\uparrow$       | NRMSE$_{\text{FULL/ROI}}$ $\downarrow$     |
> | Base          |  18.50 $\pm$ 0.36  /  18.37 $\pm$ 0.54   |  3.32 $\pm$ 0.35  /  4.85 $\pm$ 0.51 |  0.13 $\pm$ 0.01  /  0.13 $\pm$ 0.01      |
> | Less Layer    |  17.65 $\pm$ 0.80  /  18.09 $\pm$ 1.32  |  2.47 $\pm$ 0.78  /  4.56 $\pm$ 1.29 |  0.14 $\pm$ 0.01  /  0.13 $\pm$ 0.02      |
> | More Layer    |  **18.61 $\pm$ 0.16**  /  18.47 $\pm$ 0.16   |  **3.44 $\pm$ 0.14**  /  4.95 $\pm$ 0.15|  **0.12 $\pm$ 0.00**  /  **0.12 $\pm$ 0.00**      |
> | No Res Block   |  18.47 $\pm$ 0.23  /  **18.71 $\pm$ 0.62** |  3.29 $\pm$ 0.25  /  **5.18 $\pm$ 0.59** |  0.13 $\pm$ 0.00  /  0.13 $\pm$ 0.01      |
>
> | **Setting**   | **Real (NP ch1)**     |            |                      |
> |---------------|---------------------------------|---------------------------------|-----------------------------------|
> |               | PSNR$_{\text{FULL/ROI}}$ $\uparrow$        | SNDR$_{\text{FULL/ROI}}$ $\uparrow$       | NRMSE$_{\text{FULL/ROI}}$ $\downarrow$     |
> | Base          |  17.45 $\pm$ 0.39  /  16.94 $\pm$ 0.35  |  2.27 $\pm$ 0.38  /  3.67 $\pm$ 0.33     |  0.14 $\pm$ 0.01  /  0.15 $\pm$ 0.01      |
> | Less Layer    |  16.96 $\pm$ 0.38  /  16.30 $\pm$ 0.36    |  1.78 $\pm$ 0.38  /  3.02 $\pm$ 0.35     |  0.15 $\pm$ 0.01  /  0.16 $\pm$ 0.01      |
> | More Layer |  17.64 $\pm$ 0.11  /  16.91 $\pm$ 0.30  |  2.46 $\pm$ 0.06  /  3.64 $\pm$ 0.28     |  **0.14 $\pm$ 0.00**  /  **0.15 $\pm$ 0.01**      |
> | No Res Block |  **17.82 $\pm$ 0.53**  /  **17.15 $\pm$ 0.56**  |  **2.64 $\pm$ 0.50**  /  **3.88 $\pm$ 0.55**     |  0.14 $\pm$ 0.01  /  0.15 $\pm$ 0.01      |
>
>
> The decoder depth and the use of residual-block results only small changes in reconstruction metrics (less than 0.6 dB). Adding depth gives mild improvements and removing residual blocks improves ROI metrics a little, likely due to dataset noise and interaction with BatchNorm / skip connections. In all, the small magnitude of the effects shows the decoder does not dominate or mask encoder performance.
>
> We appreciate the reviewer’s careful critique; the planned additions will significantly improve clarity and reproducibility.

---

### Official Review · Reviewer_beEE · 2025-11-06

**Soundness:** 2
**Presentation:** 2
**Contribution:** 2
**Rating:** 4
**Confidence:** 3

**Summary:**

This work presents a compressor for spike-shaped biosignals with an extremely light-weight encoder, which can be deployed into an implantable sensing device, e.g., an implantable brain-machine interface (iBMI). The main innovation lies in the resource-efficient design of the encoder, which only requires 124 bits for storing the parameters and can be implemented with computationally cheap shift and add operations. Together with a new loss function that focuses on the spike reconstructions (EAL), the proposed compressor achieves Pareto-optimality with respect to reconstruction fidelity and number of encoder parameters on a series of datasets.

**Strengths:**

While being a tailored solution to the problem of compressing spike signals, the proposed compressor achieves good compression performance while being highly resource efficient on the encoder side. This angle is interesting and relevant for such sensing applications.

**Weaknesses:**

My major concern lies in the heavy tailoring of the compressor to spike-shaped biosignals. The compressor is only demonstrated on single, centered, cropped spike signals. From a practical perspective, it then seems quite hard to fully reconstruct the entire signal for analysis (e.g., classification). Obviously, this inductive bias simplifies the encoding, but makes the overall application questionable. Even more, it is not clear how and if the compressor could be applied to different kinds of time signals (even inside the biosignals domain, e.g., scalp EEG).

Minor:

-	Weak classifier baseline. Using a K-means clustering with class assignments seems rather weak. It would be good to use state-of-the-art classifiers in the domain.
-	Different time windows for experiments and hardware emulations. Experiments use a time window of 128, while the hardware emulations in B.2 reduce it to T=32. This makes the justification of the hardware difficult, as no experiments are demonstrated with the small time window.

**Questions:**

1.	How does the compressor perform in a more general setting, i.e., without cropping and centering the spikes? It would be good to have a setup with continuous streaming, where also multiple spikes could be contained inside a window.
2.	Reporting a stronger classification baseline would be appreciated.

---

> ### Author Response · Authors · 2025-11-17
> **Response to Reviewer beEE**
>
> We thank the reviewer for the time spent on reviewing this paper and the valuable comments. We have carefully read through the comments and respond to each comment as below.
>
> **Weakness:**
>
> **Comment 1:**  My major concern lies in the heavy tailoring of the compressor to spike-shaped biosignals. The compressor is only demonstrated on single, centered, cropped spike signals. From a practical perspective, it then seems quite hard to fully reconstruct the entire signal for analysis (e.g., classification). Obviously, this inductive bias simplifies the encoding, but makes the overall application questionable. Even more, it is not clear how and if the compressor could be applied to different kinds of time signals (even inside the biosignals domain, e.g., scalp EEG).
>
> **Response1:**  As pointed out by the reviewer, the proposed NSC has a simplify encoder in order to save resources on the implanted chip side. The application of the proposed NSC is for single neuron spike signal detection and compression as explained in the introduction section. And we have demonstrated the classification of the single neuron spike signal can still be done without reconstructing the entire raw signal (Table 4). The proposed NSC is suitable to process biosignal that activates within a limited time window such as single neuro spike and cardiac signal ECG. For information that embedded in a relatively long period such as LFP and EEG the proposed NSC may not be applicable. In this paper, we do have specific application scenario and target signal. Discussion of the generalization for proposed NSC has been added in the conclusion section (page. 9, line 463-476).
>
> **Comment2:**  Minor: 1. Weak classifier baseline. Using a K-means clustering with class assignments seems rather weak. It would be good to use state-of-the-art classifiers in the domain. 2. Different time windows for experiments and hardware emulations. Experiments use a time window of 128, while the hardware emulations in B.2 reduce it to T=32. This makes the justification of the hardware difficult, as no experiments are demonstrated with the small time window.
>
> **Response2:**
>
> 1. Choice of clustering baseline (K-means).
>
> We used K-means as an unsupervised classifier of whether the outputs preserve separable morphology. These complements supervised metrics but avoid bias from classifier capacity. We agree that adding supervised baselines strengthens the study, and we will include a small MLP. Results are reported in Response 4.
>
> 2. Window sizes: T=128 experiments and T=32 for hardware design.
>
> The hardware implementation is shown as a feasibility demonstration for low-buffer scenarios. The encoder's logical operations (shift/add with quantized parameters) are identical for different window lengths. In implantable AFE and digital designs with 180nm, the area of the digital block is required to be about the same scale as the analog frontend. Moving to a more advanced process node, such as 28 nm, would relieve these constraints and allow larger local buffers, such as T=128, but such process upgrades come with different trade-offs and are outside the scope of our present feasibility study. In general, this design has no change for the compression module (NSC). Mainly for the raw data storage. The parameter for the encoder remains the same as described in Table 4.

---

> ### Author Response · Authors · 2025-11-17
> **Response to Reviewer beEE**
>
> **Questions:**
>
> **Comment 3:** How does the compressor perform in a more general setting without cropping and centering the spikes for streaming and multiple spikes?
>
> **Response 3:** Current architecture for the neural signal processing cycles between detection-alignment-operation(sorting/compression)-detection-xxx. The biased spike (not centered) will mostly not happen in a real hardware implementation. The cropping and alignment will be done by the NEO detection process with stored data. The compressor can handle multiple spikes situation, as long as the time separation between two adjacent spikes is larger than 32 bits * sampling rate (for transmitting outside).
>
> And for the real streaming data testing, we have done this in VIVADO, an FPGA simulation software, with the waveform. By deploying the quantized params and inputs with the streaming data for testing the full chip design. This experiment result will also be provided in the revision.

---

> ### Author Response · Authors · 2025-11-17
> **Response to Reviewer beEE**
>
> **Comment 4:** Reporting a stronger classification baseline.
>
> **Response 4:** Here we trained two small MLP for classification with structure of 128-32-1 of the full reconstructed waveforms and 64-32-1 of the ROI reconstructed waveforms. Both are trained with the same dataset as NSC. And tested with the outputs from NSC. Lr was set to 1e-3, weight decay of 1e-4, with CELoss and AdamW optimizer 100 epochs. Same seed setting with 1-5. With train: valid of 7: 1, without test set split. All the tested data are from the reconstructed waveforms.
>
> | **Models (FULL wave)**   | AE_FP32  |  AE_INT8  |  AE_INT1P4  |  PCA  |  CS  |  VQVAE  |  BNN  |  NSC_MSE  |  NSC |
> |---------------|---------|--------------|----------|--------|----------|---------------|------------|------------|-----------|
> |**GC datasets**|
> |F1|0.89 $\pm$ 0.14  |  0.76 $\pm$ 0.12  |  0.58 $\pm$ 0.13  |  **0.99 $\pm$ 0.00**  |  0.16 $\pm$ 0.17  |  0.63 $\pm$ 0.22  |  0.30 $\pm$ 0.06  |   0.97 $\pm$ 0.04  |  0.83 $\pm$ 0.16|
> |ARI|0.86 $\pm$ 0.15  |  0.58 $\pm$ 0.18  |  0.31 $\pm$ 0.26  |  **0.99 $\pm$ 0.01**  |  0.06 $\pm$ 0.10  |  0.61 $\pm$ 0.32  |  0.00 $\pm$ 0.00  |  0.92 $\pm$ 0.11  |  0.74 $\pm$ 0.10|
> |NMI|0.81 $\pm$ 0.16  |  0.56 $\pm$ 0.14  |  0.29 $\pm$ 0.20  |  **0.97 $\pm$ 0.01**  |  0.09 $\pm$ 0.13  |  0.58 $\pm$ 0.31  |  0.00 $\pm$ 0.00  |  0.91 $\pm$ 0.09  |  0.73 $\pm$ 0.09|
> |**hc1 datasets**|
> |F1|0.99 $\pm$ 0.00  |  0.84 $\pm$ 0.06  |  0.78 $\pm$ 0.08  |  **1.00 $\pm$ 0.00**  |  0.43 $\pm$ 0.18  |  0.96 $\pm$ 0.01  |  0.92 $\pm$ 0.03  |  0.96 $\pm$ 0.01  |  0.96 $\pm$ 0.03|
> |ARI|0.95 $\pm$ 0.02  |  0.49 $\pm$ 0.15  |  0.34 $\pm$ 0.17  |  **0.99 $\pm$ 0.00**  |  0.03 $\pm$ 0.04  |  0.85 $\pm$ 0.04  |  0.71 $\pm$ 0.10  |  0.85 $\pm$ 0.02  |  0.84 $\pm$ 0.10|
> |NMI|0.90 $\pm$ 0.03  |  0.39 $\pm$ 0.13  |  0.27 $\pm$ 0.14  |  **0.97 $\pm$ 0.01**  |  0.03 $\pm$ 0.03  |  0.77 $\pm$ 0.05  |  0.62 $\pm$ 0.07  |  0.76 $\pm$ 0.03  |  0.77 $\pm$ 0.10|
>
>
>
> | **Models (ROI wave)**   | AE_FP32  |  AE_INT8  |  AE_INT1P4  |  PCA  |  CS  |  VQVAE  |  BNN  |  NSC_MSE  |  NSC |
> |---------------|---------|--------------|----------|--------|----------|---------------|------------|------------|-----------|
> |**GC datasets**|
> |F1|0.98 $\pm$ 0.01  |  0.76 $\pm$ 0.13  |  0.46 $\pm$ 0.29  |  **1.00 $\pm$ 0.00**  |  0.13 $\pm$ 0.12  |  0.75 $\pm$ 0.26  |  0.33 $\pm$ 0.01  |  0.97 $\pm$ 0.05  |  **1.00 $\pm$ 0.00**|
> |ARI|0.96 $\pm$ 0.03  |  0.50 $\pm$ 0.24  |  0.31 $\pm$ 0.27  | **0.99 $\pm$ 0.00**  |  0.00 $\pm$ 0.00  |  0.67 $\pm$ 0.35  |  0.00 $\pm$ 0.00  |  0.91 $\pm$ 0.12  |  **0.99 $\pm$ 0.01**|
> |NMI|0.92 $\pm$ 0.04  |  0.47 $\pm$ 0.19  |  0.26 $\pm$ 0.23  |  **0.98 $\pm$ 0.00**  |  0.01 $\pm$ 0.01  |  0.62 $\pm$ 0.33  |  0.00 $\pm$ 0.00  |  0.90 $\pm$ 0.09  |  0.97 $\pm$ 0.01|
> |**hc1 datasets**|
> |F1|0.99 $\pm$ 0.00  |  0.83 $\pm$ 0.07  |  0.77 $\pm$ 0.08  |  **1.00 $\pm$ 0.00**  |  0.53 $\pm$ 0.18  |  0.96 $\pm$ 0.01  |  0.92 $\pm$ 0.02  |  0.96 $\pm$ 0.01  |  0.97 $\pm$ 0.01|
> |ARI|0.96 $\pm$ 0.01  |  0.47 $\pm$ 0.17  |  0.33 $\pm$ 0.18  |  **0.99 $\pm$ 0.00**  |  0.08 $\pm$ 0.15  |  0.85 $\pm$ 0.05  |  0.71 $\pm$ 0.07  |  0.86 $\pm$ 0.03  |  0.89 $\pm$ 0.02|
> |NMI|0.91 $\pm$ 0.02  |  0.38 $\pm$ 0.13  |  0.26 $\pm$ 0.15  | **0.97 $\pm$ 0.01**  |  0.11 $\pm$ 0.12  |  0.76 $\pm$ 0.06  |  0.61 $\pm$ 0.05  |  0.78 $\pm$ 0.04  |  0.81 $\pm$ 0.03||ARI|0.96 $\pm$ 0.01  |  0.47 $\pm$ 0.17  |  0.33 $\pm$ 0.18  |  0.99 $\pm$ 0.00  |  0.08 $\pm$ 0.15  |  0.85 $\pm$ 0.05  |  0.71 $\pm$ 0.07  |  0.86 $\pm$ 0.03  |  0.89 $\pm$ 0.02|
> |NMI|0.91 $\pm$ 0.02  |  0.38 $\pm$ 0.13  |  0.26 $\pm$ 0.15  |  0.97 $\pm$ 0.01  |  0.11 $\pm$ 0.12  |  0.76 $\pm$ 0.06  |  0.61 $\pm$ 0.05  |  0.78 $\pm$ 0.04  |  0.81 $\pm$ 0.03|
>
> NSC (with EAL) achieves high ROI performance in the supervised evaluation, catching up AE_FP32 while using much fewer encoder resources. This indicates that NSC preserves the spike morphology under extreme hardware constraints. On full-wave classification, PCA shows very highest scores cross both datasets. However, PCA relies on dense matrix multiplications and large floating-point projection matrix, making it infeasible for implant deployment. Moreover NSC maintains strong downstream clustering performance while remaining within a realistic implant resource constrain.
>
> We appreciate the reviewer’s suggestions — they will improve clarity and practical relevance. We will add the mentioned results in the revision.

---

> > ### Comment · Reviewer_beEE · 2025-11-25
> >
> > I would like to thank the authors for their detailed response and for conducting additional experiments during the rebuttal period.
> >
> > While I appreciate the clarification regarding the hardware pipeline and the addition of the MLP results, my reservations regarding the generality and the comparative baselines remain.
> >
> > 1. **Generality and Robustness:** The authors explain that the "detection-alignment" cycle ensures the compressor only receives centered data. However, strictly decoupling detection from compression relies on the assumption of perfect upstream detection. In practical in-vivo settings, perfect alignment is rarely guaranteed. The lack of experiments demonstrating the compressor's robustness to non-ideal inputs (e.g., jittered alignment, overlapping spikes, or artifacts) limits the assessment of its reliability in a real-world iBMI pipeline. I believe characterizing this failure mode is crucial for a specialized hardware accelerator.
> > 2. **Classification Baselines:** Regarding the new supervised baselines, while the addition of an MLP is an improvement over K-Means, it does not necessarily represent the state-of-the-art for time-series classification. Given the sequence nature of the data, models utilizing 1D-Convolutions (1D-CNN), recurrence (RNN/LSTM) or attention (Transformer) are standard benchmarks that often outperform simple MLPs on waveform data.

---

> ### Author Response · Authors · 2025-11-28
> **Second Response to Reviewer beEE**
>
> Thanks for your response and comments, and we will respond to each comment as follows.
>
> **Comment 1:**  Generality and Robustness: The authors explain that the "detection-alignment" cycle ensures the compressor only receives centered data. However, strictly decoupling detection from compression relies on the assumption of perfect upstream detection. In practical in-vivo settings, perfect alignment is rarely guaranteed. The lack of experiments demonstrating the compressor's robustness to non-ideal inputs (e.g., jittered alignment, overlapping spikes, or artifacts) limits the assessment of its reliability in a real-world iBMI pipeline. I believe characterizing this failure mode is crucial for a specialized hardware accelerator.
>
> **Response1:**  (1) Design Philosophy. Our central contribution is a hardware-efficient compression encoder designed for severe area/power constraints. Prior research on on-chip compression, spike detection, and alignment can be treated as upstream tasks. This separation allows us to focus on minimizing transmitted bits without duplicating the complexity of the frontend signal processing chain.
>
> (2) Practical Mitigation. We agree that robustness is critical in deployed systems. However, in practical ASIC (hardware) design, handle jitter and artifacts via various upstream strategies rather than the encoder itself. The approaches of adaptive thresholds, refractory windows, debounce and padding all effectively mitigate false triggers and overlaps with negligible digital area overhead. These upstream modules effectively preprocess the input, ensuring the encoder operates within its design specifications.
>
> **Comment2:**  Classification Baselines: Regarding the new supervised baselines, while the addition of an MLP is an improvement over K-Means, it does not necessarily represent the state-of-the-art for time-series classification. Given the sequence nature of the data, models utilizing 1D-Convolutions (1D-CNN), recurrence (RNN/LSTM) or attention (Transformer) are standard benchmarks that often outperform simple MLPs on waveform data.
>
> **Response2:** We thank the reviewer for the suggestion to include stronger sequence models. Here are the results for classification with the transformer model, of d_model=32, n_head = 4, num_layer=2, with the same settings as before, which will be reported in revision. The results on the GC datasets are reported below; hc1 will report in the revision.
>
> | **Models (FULL wave)**   | AE_FP32  |  AE_INT8  |  AE_INT1P4  |  PCA  |  CS  |  VQVAE  |  BNN  |  NSC_MSE  |  NSC |
> |---------------|---------|--------------|----------|--------|----------|---------------|------------|------------|-----------|
> |F1|0.91 $\pm$ 0.14 | 0.71 $\pm$ 0.21 | 0.47 $\pm$ 0.23 | **0.97 $\pm$ 0.06** | 0.04 $\pm$ 0.00 | 0.46 $\pm$ 0.29 | 0.19 $\pm$ 0.01 | 0.74 $\pm$ 0.15 | 0.85 $\pm$ 0.25|
> |ARI|0.88 $\pm$ 0.17 | 0.48 $\pm$ 0.23 | 0.23 $\pm$ 0.27 | **0.93 $\pm$ 0.13** | 0.00 $\pm$ 0.00 | 0.31 $\pm$ 0.33 | 0.00 $\pm$ 0.00 | 0.38 $\pm$ 0.22 | 0.79 $\pm$ 0.24|
> |NMI|0.88 $\pm$ 0.12 | 0.44 $\pm$ 0.19 | 0.24 $\pm$ 0.21 | **0.94 $\pm$ 0.10** | 0.00 $\pm$ 0.00 | 0.31 $\pm$ 0.32 | 0.00 $\pm$ 0.00 | 0.49 $\pm$ 0.20 | 0.84 $\pm$ 0.15|
>
>
> | **Models (ROI wave)**   | AE_FP32  |  AE_INT8  |  AE_INT1P4  |  PCA  |  CS  |  VQVAE  |  BNN  |  NSC_MSE  |  NSC |
> |---------------|---------|--------------|----------|--------|----------|---------------|------------|------------|-----------|
> |F1|0.99 $\pm$ 0.01 | 0.82 $\pm$ 0.07 | 0.56 $\pm$ 0.21 | **1.00 $\pm$ 0.00** | 0.04 $\pm$ 0.00 | 0.68 $\pm$ 0.33 | 0.31 $\pm$ 0.06 | 0.87 $\pm$ 0.09 | **1.00 $\pm$ 0.00**|
> |ARI|0.97 $\pm$ 0.02 | 0.61 $\pm$ 0.15 | 0.29 $\pm$ 0.26 | **1.00 $\pm$ 0.00** | 0.00 $\pm$ 0.00 | 0.52 $\pm$ 0.44 | -0.00 $\pm$ 0.00 | 0.64 $\pm$ 0.22 | 0.99 $\pm$ 0.01|
> |NMI|0.95 $\pm$ 0.03 | 0.56 $\pm$ 0.16 | 0.25 $\pm$ 0.21 | **0.99 $\pm$ 0.01** | 0.00 $\pm$ 0.00 | 0.49 $\pm$ 0.40 | 0.00 $\pm$ 0.00 | 0.68 $\pm$ 0.17 | 0.98 $\pm$ 0.01|
>
> These experiments confirm that NSC remains competitive with baselines on ROI fidelity while maintaining the strict hardware efficiency.

---

### Meta-Review · Area_Chair_Pv2b · 2025-12-27

**Summary:**

This paper is a highly specialized study that designs a neural codec for brain signals. The primary contribution is the introduction of a resource-efficient encoder that uses only 124 bits for storage, suitable for a mobile edge device. In addition, the Energy-Aware Loss (EAL) function is proposed to optimize the signal fidelity, especially the reconstruction of high-energy spike regions over background noise.

**Reviewer Concerns:**

The reviewers' primary concerns center on the system's practical robustness, especially how the encoder would handle real-world signals that can be messy, unaligned, or overlapping. Since the experiments relied on perfectly centered spikes, these concerns are unresolved. They also criticized the fairness of the initial comparisons and the use of weak classification baselines, pointing out that the results did not clearly demonstrate if the performance trade-off was acceptable for biological analysis.  There were also complaints about the presentation, specifically the low-quality figures and the lack of explanations of the decoder architecture.

The authors addressed these concerns by providing new classification results using MLP and Transformer models, demonstrating that the NSC effectively preserves biological features. They also retrained baseline models with the EAL loss to show that their architectural efficiency.

Ultimately, the paper is considered borderline, offering significant hardware advantages while leaving some questions about real-world robustness outstanding.

**Reviewer Scores:**

Reviewer beEE would keep 4 because their final feedback expressed persistent concerns about the assumption of perfect spike alignment and its inability to handle real-world artifacts.

Reviewer TXDs would also likely stay at a marginal score, as they felt the design choices and the "double smoothing" in the loss function remained poorly motivated despite the promised figure updates.

While Reviewer AAjh was unconvinced by the additional experiments.

The lack of any reviewer moving into the "Accept" range (6+) suggests that they reach a consensus that the work, while technically interesting, is too narrow in scope and insufficiently robust for a general machine learning audience.

---

### Decision · Program_Chairs · 2026-01-26

Reject